# An entanglement association polymer electrolyte for Li-metal batteries

Hangchao Wang[1], Yali Yang[1], Chuan Gao[1], Tao Chen[1], Jin Song[1], Yuxuan Zuo[1], Qiu Fang[2], Tonghuan Yang[1], Wukun Xiao[1], Kun Zhang[1], Xuefeng Wang [2,3,4] ✉ & Dingguo Xia [1,2] ✉

To improve the interface stability between Li-rich Mn-based oxide cathodes and electrolytes, it is necessary to develop new polymer electrolytes. Here, we report an entanglement association polymer electrolyte (PVFH-PVCA) based on a poly (vinylidene fluoride-co-hexafluoropropylene) (PVFH) matrix and a copolymer stabilizer (PVCA) prepared from acrylonitrile, maleic anhydride, and vinylene carbonate. The entangled structure of the PVFH-PVCA electrolyte imparts excellent mechanical properties and eliminates the stress arising from dendrite growth during cycling and forms a stable interface layer, enabling Li// Li symmetric cells to cycle steadily for more than 4500 h at 8 mA cm$^{-2}$. The PVCA acts as a stabilizer to promote the formation of an electrochemically robust cathode−electrolyte interphase. It delivers a high specific capacity and excellent cycling stability with 84.7% capacity retention after 400 cycles. $Li_{1.2}Mn_{0.56}Ni_{0.16}Co_{0.08}O_2$/PVFH-PVCA/Li full cell achieved 125 cycles at 1 C (4.8 V cut-off) with a stable discharge capacity of ~2.5 mAh cm$^{-2}$.

The development of electric vehicles, energy storage, and consumer electronics industries has driven global demand for high-energy-density batteries[1]. Compared with Li-ion batteries, Li-metal batteries (LMBs) based on liquid electrolytes have a higher energy density[2], but internal short circuit caused by lithium dendrite growth poses serious safety hazards[3,4]. On the other hand, LMBs based on solid electrolytes have high energy density and high safety[5–7] but suffer from issues such as low ion conductivity and poor interface contact[8–10]. Gel polymer electrolytes have attracted considerable attention owing to their improved safety[11], reduced interface impedance between the solid electrolyte and electrode[12], and compatibility with the current roll-to-roll LMB manufacturing process[13,14].

Among the polymer-based gel electrolytes, polyacrylonitrile has a wide electrochemical window and high ionic conductivity[15] but poor compatibility with the Li-metal anode[16]. Although PVFH has inadequate mechanical properties, it boasts a wider electrochemical window and higher liquid absorption rate[17]. Additionally, anhydride oxidation can facilitate the formation of a stable passivation film on the cathode

surface, effectively mitigating issues such as electrolyte decomposition, transition metal dissolution, and cathodic erosion[18]. However, gel electrolytes based on a single polymer substrate have certain critical drawbacks that render them unsuitable for practical applications. In the pursuit of high-performance polymers, it has been consistently challenging to simultaneously optimize both the electrochemical and mechanical properties of gel polymers.

A stable entanglement structure, formed by the intermolecular interactions of surrounding molecular chains, plays a crucial role in dictating the macroscopic mechanical properties of polymer materials[19,20]. Block copolymers offer the advantage of combined excellent properties derived from different monomers, allowing control over the relative molecular weight, molecular structure, and composition[21–23]. When used in polymer electrolytes, block copolymers with multiple polar groups synergistically enhance the electrochemical properties, such as ionic conductivity, while promoting entanglement between the amorphous polymer chains, thereby ensuring desirable mechanical properties[24–26]. Thus, we envisioned an approach of blending

[1]Beijing Key Laboratory of Theory and Technology for Advanced Batteries Materials, School of Materials Science and Engineering, Peking University, Beijing 100871, China. [2]Institute of carbon neutrality, Peking University, Beijing 100871, China. [3]Laboratory for Advanced Materials & Electron Microscopy, Institute of Physics, Chinese Academy of Sciences, Beijing 100190, China. [4]College of Materials Science and Opto-Electronic Technology, University of Chinese Academy of Sciences, Beijing 100049, China. ✉e-mail: wxf@iphy.ac.cn; dgxia@pku.edu.cn

triblock polymers with diverse functional groups with PVFH, which has excellent electrochemical stability. Strong interactions between the two polymers can facilitate the formation of a blend characterized by a high-density chain entanglement structure and a mechanically robust homogeneous dense matrix. This polymer blend design aims to achieve exceptional ion conduction and mechanical properties.

In this work, we successfully synthesized a novel triblock copolymer, specifically, a copolymer stabilizer (denoted as PVCA) comprising acrylonitrile (AN), maleic anhydride (MA), and vinylene carbonate (VC), and blended it with PVFH to develop a new high-voltage gel polymer electrolyte (denoted as PVFH-PVCA). Owing to the combined functional properties of the nitrile, ester, and anhydride groups, this system has a high ion conductivity and strong antioxidant potential. Importantly, the robust polar groups in PVCA interact with PVFH, facilitating the formation of a high-density entangled structure, which effectively improves the mechanical properties of the gel polymer electrolyte, leading to an impressive tensile strength of 19.83 MPa. The PVFH-PVCA electrolyte-based battery assembled with Li-rich Mn-based materials as the cathode and Li metal as the anode exhibits a long cycle life and high discharge capacity. Similar improvements were also realized in LiCoO$_2$ (LCO)||Li, LiFePO$_4$ (LFP)||Li, and LiNi$_{0.95}$Co$_{0.05}$O$_2$ (NC95)||Li cells, which prove the universal applicability of PVFH-PVCA as a prospective polymer electrolyte.

## Results and discussion
### Synthesis and characterization of PVCA
Figure 1a depicts the operational mechanism of quasi-solid-state LMBs produced using the strategically designed PVFH-PVCA electrolyte.

Conventional liquid electrolyte (LE)-based Li-ion batteries suffer from issues, such as unstable solid electrolyte interface (SEI) formation and electrolyte decomposition, resulting in dendrite growth and cathode degradation (Fig. 1a, left)[27,28]. The PVFH-PVCA electrolyte developed in this study can generate a stable SEI on the Li-metal anode, promoting uniform Li deposition. Simultaneously, PVCA participates in the formation of the cathode–electrolyte interphase (CEI), suppressing electrolyte decomposition and ensuring the structural stability of the LRMO (Fig. 1a, right). To prepare the PVCA ternary copolymer via free-radical polymerization, and introduce PVCA into PVFH to form a dense polymer film, and the corresponding synthetic scheme is shown in Supplementary Fig. S1. A tailor-made multifunctional macromolecule, PVCA, was synthesized via a one-step thermally initiated copolymerization using MA, VC, and AN as monomers. Supplementary Fig. S2 shows the Fourier transform infrared (FTIR) spectrum of the PVCA copolymer. The characteristic absorption peaks of the methine (-CH) group stretching vibrations of the AN at 3165, 3126, and 3064 cm$^{-1}$ disappeared and that of the -CN group appeared at 2236 cm$^{-1}$. Further, the characteristic stretching vibration peaks of the C=O groups in MA and VC appeared at 1778 and 1341 cm$^{-1}$. The synthesis of PVCA was confirmed by the absence of the unsaturated C=C moieties in the FTIR spectrum owing to the successful random copolymerization of AN, MA, and VC. Furthermore, the $^1$H-NMR spectrum of PVCA was analyzed and the peaks in the spectrum were attributed (Supplementary Fig. 3a). It can be found that the distinct peaks appeared around the low field (2-3.5 ppm), which can indicate the appearance of hydrogen atoms connected by C-C single bonds. Additionally, the integration results of the peaks show that the ratio of the number of substrates involved in

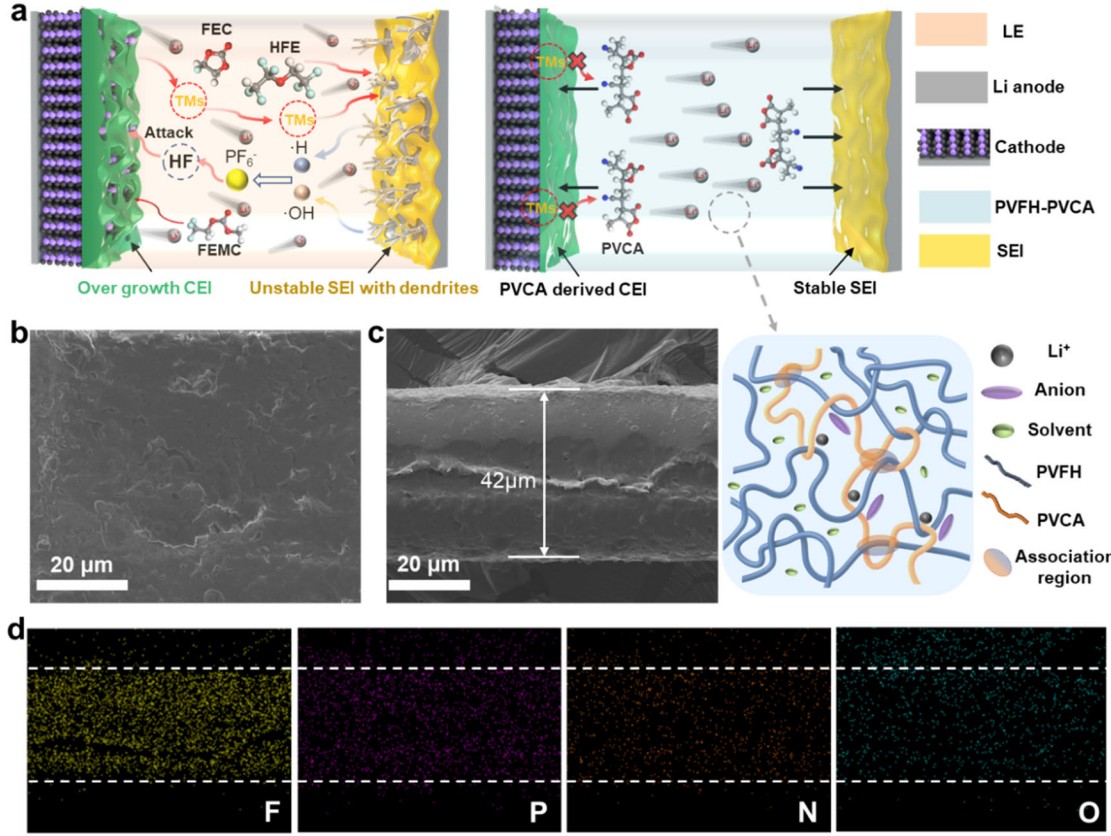

**Fig. 1 | Design of PVFH-PVCA polymer electrolytes for high-performance Li-metal batteries. a** Schematic illustration of the dendrites formed on the Li-metal electrode and degradation processes occurring in cathodes upon using a carbonate-based electrolyte solution (left) and schematic illustration of the stable solid electrolyte interphase (SEI) formed on the Li-metal anode and the inhibited degradation of the cathode interface layer in a Li-metal cell using the PVFH-PVCA polymer electrolyte developed in this study (right). **b** and **c** SEM images of the surface **b** and cross-section **c** of the PVFH-PVCA polymer electrolyte. **d** Energy-dispersive spectroscopic elemental mapping images of F, P, N, and O in the PVFH-PVCA gel polymer electrolyte.

the reaction in PVAC is MA:VC:AN = 1:3:3. By analyzing the ¹³C-NMR spectra of the reactants MA, VC, AN, and PVAC (Supplementary Fig. 3b–e), it can be observed that the peaks of C=C double bonds (100–140 ppm) can be observed in the reactants MA, VC, and AN. The peak intensity of C=C double clicking in the product PVAC is significantly reduced, and there is a clear C-C single bond peak (20-40ppm) in the low field, confirming that the substrate did undergo polymerization reaction. As shown in Supplementary Fig. 4a, the molecular weight of PVCA was determined to be 91,100 using Gel Permeation Chromatography (GPC). Thermogravimetric analysis revealed that the copolymer exhibits a single decomposition temperature (Supplementary Fig. S4b).

### Structure and morphology of PVFH-PVCA blends

To demonstrate the potential of PVCA as a polymer electrolyte component of high-energy-density LMBs, we selected PVFH, an electrochemically stable polymer that is commonly used in quasi-solid polymer electrolytes, as the polymer matrix. Dynamic mechanical analysis revealed that PVFH-PVCA membranes with 15 and 20 wt% PVCA have better mechanical properties (Supplementary Fig. S5) than

others. The highest conductivity of 2.04 × 10⁻³ S cm⁻¹ at 25 °C was achieved for the sample with 15 wt% PVCA (Supplementary Fig. S6). Therefore, a PVCA content of 15 wt% was used for further analyses.

Figure 2d shows the FT-IR spectra of LE-free PVCA, PVFH, and PVFH-PVCA. The FTIR spectrum of PVFH presented peaks at 1166 and 1227 cm⁻¹, corresponding to C-F stretching vibrations and -CF₂- bending vibrations, respectively. For LE-free PVFH-PVCA, the stretching vibration peak of -CF₂- (1227 cm⁻¹) shifted to a higher wavenumber while that of the -C = O of the carbonate moiety shifted to a lower wavenumber. Further, a new broad peak appeared at 3415 cm⁻¹, partly due to the dipole–dipole interaction between the -C = O groups of the PVCA skeleton and the C-F groups of PVFH, which promoted compatibility between the two polymers[29,30]. On the other hand, the -CN group increases the electronegativity of the carbon atom of the methine group, favoring strong H⋯F interactions between the -CH- and -CF₂- groups in PVFH, resulting in a strong H⋯F stretching vibration peak[31].

### Interaction between the blended polymers and lithium salt

Upon soaking with the LE, the dry light-yellow PVFH-PVCA membrane gradually turned into a transparent gel at room temperature

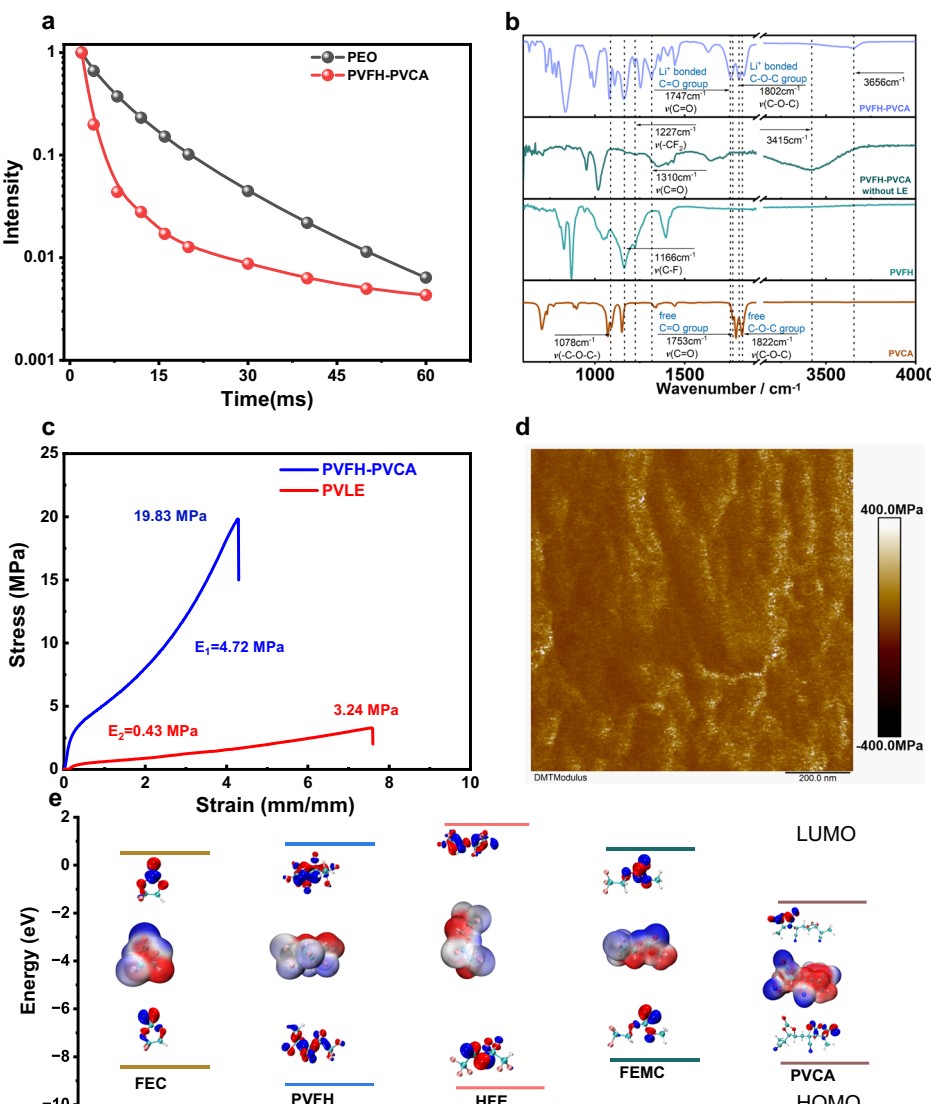

**Fig. 2 | Characterization of the PVFH-PVCA polymer electrolyte. a** ¹H-NMR T₂ relaxation curves for PVFH-PVCA and PEO. Adsorption energy of PVFH with PVCA (− 44.461 kJ/mol). **b** FTIR spectra of PVCA, PVFH, and PVFH-PVCA without LE and PVFH-PVCA recorded at 25 °C. **c** Stress−strain curves of PVFH-PVCA and PVLE at an extension rate of 100 mm min⁻¹. **d** Shear modulus of PVFH-PVCA. **e** LUMO and HOMO energy values of the solvent molecules (FEC, HFE, and FEMC) and polymers (PVFH and PVCA). Insets show the molecular electrostatic potentials and the corresponding visual LUMO and HOMO geometrical structures.

(Supplementary Fig. S1b). Accordingly, the stretching vibration of H⋯F shifted to a higher wavenumber (Fig. 2b, top) owing to the formation of hydrogen bonds between the polymer skeleton and fluoroethylene carbonate (FEC) and methyl 2,2,2-trifluoroethyl carbonate (FEMC) in the LE. The high-density H-bonds between the membrane and electrolyte, combined with the H-bonds between FEC and FEMC and between PVFH and PVCA, cause the synergistic crosslinking of the two polymers at multiple sites, significantly improving the mechanical strength of the gel electrolyte. Moreover, the C-O-C and C=O groups coordinate with Li$^+$, as indicated by the shift in their characteristic stretching vibrations to lower wavenumbers[32]. In addition, the polar -CN functional groups in the polymer chain can also interact with Li$^+$, providing a fast Li$^+$-diffusion channel, thereby regulating Li$^+$ distribution during lithium deposition/stripping processes, resulting in high Li$^+$ conductivity[33].

PVFH soaked in the LE (PVLE) and Celgard with 60 µL of the LE (CLE) were selected as control samples. PVLE has a loose, porous surface with a thickness of 47 µm (Supplementary Fig. S7 and S8), whereas the PVFH-PVCA membrane has a notably flatter and more compact structure attributable to stronger interactions between PVCA and PVFH (see Fig. 1b, c and Supplementary Fig. S9), resulting in a high adsorption energy (Supplementary Fig. S10). Moreover, both PVCA and LE were evenly distributed in the PVFH matrix (see Fig. 1d). Furthermore, the LE content in the PVFH-PVCA was ~13.7 wt% (Supplementary Fig. S11); thus, the high ionic conductivity of PVFH-PVCA is not solely due to the LE. To further investigate ion coordination in the gel electrolyte, we compared the structural characteristics of PVLE and PVFH-PVCA using Raman spectroscopy. In Li-salt solutions, cation−anion clusters can be categorized as free anion clusters (FA), loosely bound ion pairs (LIP), and aggregated clusters (AC)[34–36]. As shown in Supplementary Fig. S12, the FA, LIP, and AC contents of PVLE are 40.9, 36.3, and 22.8%, respectively, while those of PVFH-PVCA are higher at 57.1, 33.2, and 9.7%, respectively. Thus, PVCA activates the dissociation of Li salts, leading to a greater number of mobile Li$^+$.

## Mechanical and electrochemical properties of PVFH-PVCA

The $T_2$ relaxation behaviors of different polymers were assessed using $^1$H Hahn echo, obtaining relaxation curves of the signal intensity as a function of time on a logarithmic scale (Fig. 2a). To further validate the entangled structure of PVFH-PVCA, high-molecular-weight polyethylene oxide (PEO) samples with characteristic chain entanglement were selected as reference samples[37]. The results revealed a pronounced curvature in the relaxation curve of PEO, indicating the presence of multiple or even non-exponential relaxation components[38]. Similarly, the relaxation curve of PVFH-PVCA indicated rapid decay times (less than ~20 ms), corresponding to the motion of entangled chains within the system, and slow decay times (greater than ~30 ms), representing the motion of chain ends and short chains within the system. Thus, it can be inferred that PVFH-PVCA not only shortens the $T_2$ relaxation time but also transforms the single-exponential decay behavior to a biexponential or even non-exponential one[37,39,40].

A freestanding polymer electrolyte membrane with high flexibility and mechanical strength is highly desirable for improving compatibility with electrodes and also for ensuring safety during operation. Entangled association of polymer chains can effectively enhance the mechanical properties of polymer membranes. The PVFH-PVCA membrane exhibited a significantly higher fracture toughness (19.83 MPa) than that of PVLE (3.24 MPa). In addition, the tensile curve indicated that PVFH-PVCA exhibits a high entanglement phenomenon[20]. Notably, PVFH-PVCA has a much larger elastic modulus of 4.72 MPa than PVLE (0.43 MPa; Fig. 2c). Furthermore, the PVFH-PVCA membrane could be stretched to ~4 times its original length without being ruptured (Supplementary Fig. S13). This remarkable feature can be attributed to the dynamic and reversible fracture and recombination of high-density H-bonds under external forces, which effectively dissipate energy and impart the PVFH-PVCA film with exceptional toughness. Atomic force microscopy (AFM) investigations revealed that the PVFH-PVCA membrane has a significantly lower surface roughness and a much larger Young's modulus(188.5 MPa) than PVLE (26.5 MPa, Fig. 2d and Supplementary Fig. S14). High strength is favorable for inhibiting the growth of Li dendrites, while high strain is essential for large-scale fabrication. The mechanism underlying these prominent changes are possibly the presence of certain stiff groups in PVCA, which assemble into a robust mechanical phase.

Control over the compositions of the SEI and CEI requires that the preferred interfacial species should have lower LUMO energies and higher HOMO energies than the main components[41–43]. As shown in Fig. 2e, PVCA has the lowest LUMO energy level and a relatively high HOMO energy level, indicating its ability to preferentially participate in the formation of SEI and CEI. A wide electrochemical stability window of a polymer electrolyte is crucial for enhancing the energy density of LMBs. Linear sweep voltammetry (LSV) was performed to monitor the electrochemical stabilities of PVLE, CLE, and PVFH-PVCA gel electrolytes. According to the LSV curves in Fig. 3a, the oxidation potentials of PVLE and CLE are similar at ~4.7 V vs. Li$^+$/Li, while that of PVFH-PVCA is higher (~5.3 V). The increased oxidation potential indicates the suitability of PVFH-PVCA for high-voltage cathodes. The PVFH-PVCA gel electrolyte exhibited a significantly higher ionic conductivity at 25 °C ($2.04 \times 10^{-3}$ S cm$^{-1}$) than that of PVLE ($0.94 \times 10^{-3}$ S cm$^{-1}$; Fig. 3b). Furthermore, it presented a higher Li$^+$-transference number (0.61) than those of CLE (0.34) and PVLE (0.31) (Fig. 3c and Supplementary Fig. S15), indicating that the migration of anions in PVFH-PVCA was significantly inhibited. On the one hand, a rich diversity of Li$^+$ coordination (C = O⋯Li$^+$, -CN⋯Li$^+$, and C-O⋯Li$^+$) can increase the Li$^+$ conductivity, as shown in Supplementary Fig. S16. On the other hand, the entangled association of the two polymers in PVFH-PVCA contributes to the formation of a complex physical crosslinked network, which could considerably impede anion migration.

The premise of LMB application is achieving a high lithium plating/stripping Coulombic efficiency. The PVFH-PVCA gel electrolyte not only facilitates the lowest polarization between Li stripping and plating processes (Fig. 3e) but also displays a higher and more stable Li Coulombic efficiency over cycling (Fig. 3d), indicating an improved interfacial compatibility with the Li anode. This behavior can be attributed to the uniform plating and stripping of the Li metal in each cycle while the excellent ion-transport properties can effectively alleviate electrode polarization and stabilize the SEI on the Li anode[6].

## Li plating and stripping behavior

To validate the hypothesis, the morphology of electrodeposited Li metal was characterized by Scanning electron microscopy (SEM). As shown in Fig. 4a, d, the Li| PVLE | Cu cell exhibited a highly loose and porous deposition structure, with a thickness of 49.7 µm, far exceeding the theoretical thickness value (i.e., 19.4 µm). Porous and fibrous Li metal was observed with the CLE (Fig. 4b, e). In contrast, in the cell using PVFH-PVCA electrolyte, a dendrite-free plating structure with compactly aggregated bulk material was observed (Fig. 4c). The thickness of the Li layer deposited in PVFH-PVCA electrolyte was only approximately 21.1µm (Fig. 4f), which was very close to the theoretically expected value. These results indicate that PVFH-PVCA can deposit Li$^+$ uniformly to form a dense Li metal layer (Fig. 4g). Such a dense Li deposition with a smaller surface/volume ratio effectively minimizes the parasitic reaction between metallic Li and electrolyte[44], and thus enables the high Coulombic efficiency of Li| PVFH-PVCA | Cu cells (Fig. 3d).

To investigate the compatibility between the PVFH-PVCA electrolyte and Li metal, we evaluated symmetric Li||Li cells at 30 °C (Fig. 5a). At a current density of 8 mA cm$^{-2}$ (capacity: 4 mAh cm$^{-2}$), the voltage curve of the cell with the PVLE showed a sustained increase in

 

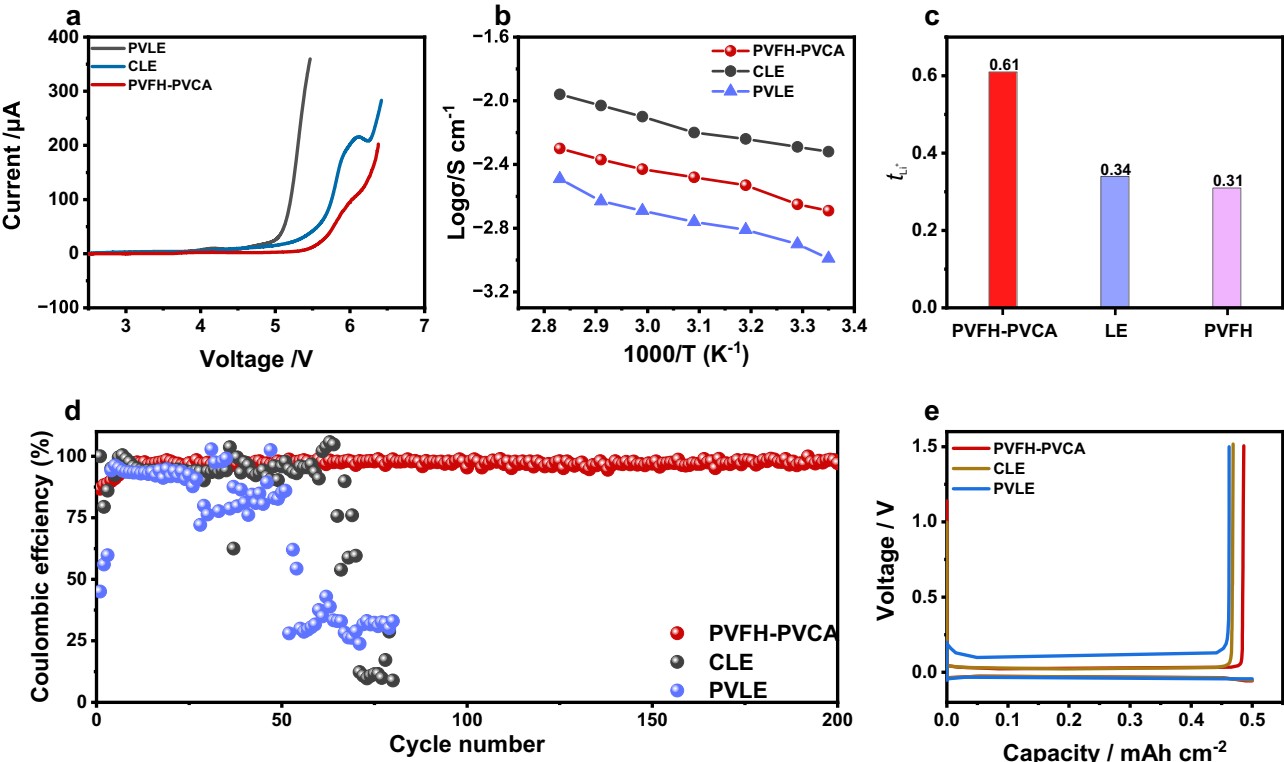

**Fig. 3 | Properties of the PVFH-PVCA electrolyte. a** Electrochemical stability window of PVFH-PVCA, CLE, and PVLE evaluated by linear sweep voltammetry. **b** Arrhenius plots of PVFH-PVCA and its components (CLE and PVLE) based on ionic conductivity ($\sigma$) versus temperature (T). **c** Comparison of the Li$^+$-transference numbers of PVFH-PVCA, CLE, and PVLE. **d** Lithium Coulombic efficiency measured in Li||Cu cells with PVFH-PVCA, CLE, and PVLE. **e** Voltage profiles of lithium plating/stripping in the Li||Cu cells with PVFH-PVCA, CLE, and PVLE at 0.5 mA cm$^{-2}$.

polarization, and the cell with CLE failed after ~1100 h because of the formation of an unstable SEI and Li dendrite growth. In contrast, the cell with the designed PVFH-PVCA electrolyte could be stably cycled for more than 4500 h, and the flat voltage plateau during plating/stripping remained steady throughout the long-term cycling period. The Li||Li cell with the PVFH-PVCA electrolyte could be operated for more than 1400 h even at a higher capacity of 5 mAh cm$^{-2}$ (10 mA cm$^{-2}$) (Supplementary Fig. S17). Remarkably, no arch shape caused by Li dendrites and dead Li accumulation was observed at the edge of the voltage profile. Figure 5b displays the voltage profiles of symmetric Li||Li cells with PVFH-PVCA at various current densities in the 1–10 mA cm$^{-2}$ range. Owing to the high ionic conductivity and excellent interfacial compatibility of the PVFH-PVCA electrolyte, the overpotential of the symmetric Li||Li cell at 10 mA cm$^{-2}$ was sustained at ~32 mV. In contrast, the PVLE cell was overpolarized at 8 mA cm$^{-2}$, and the CLE cell failed (Supplementary Fig. S18).

The morphologies of the Li-metal anodes in cells with different electrolytes were examined to gain insights into SEI formation and the microstructural evolution of the electrodes after 20 cycles at 8 mA cm$^{-2}$. The surface of the cycled Li anode of the cell with PVFH-PVCA was smooth without black deposits, and the SEM images exhibited large granular, uniform, and compact structures (Supplementary Fig. S19a and d). In sharp contrast, porous whisker-like Li deposits were observed in the SEM images of the cycled Li anodes of the cells with CLE and PVLE (Supplementary Fig. S19b and c), and the Li anode surfaces were covered with black deposits (Supplementary Fig. S19e and f). The less-compact and less-active layer formed by "dead" Li, the thick SEI layer, and the porous morphology, together with the depleted Li inventory and electrolyte, lead to impedance growth and premature cell failure at the anode side[45].

A high-resolution X-ray photoelectron spectroscopy (XPS) study of the three electrolytes and Li anode interface further confirmed the stability of the Li interface in the cell with the PVFH-PVCA electrolyte (Fig. 5c and Supplementary Fig. S20, 21). The Li *1 s* spectrum of the CLE clearly presented three components centered at ~57.0, ~55.8, and ~54.4 eV, which are assigned to LiF, Li$_2$CO$_3$, and LiCOOR, respectively (where R represents alkyl groups)[6,46]. In comparison, no characteristic peak of LiCOOR appeared in the spectrum of PVLE. In the Li *1 s* spectrum of PVFH-PVCA, the Li$_2$CO$_3$ and LiCOOR peaks were not detected; instead, two peaks appeared at ~55.0 and 58.9 eV, which are assigned to N-containing species, Li$_x$N and Li-N-C[13], originating from PVCA. The presence of Li-N-O and Li$_x$N was revealed by the N *1 s* spectrum, with peaks at ~401.7 and ~398.4 eV, respectively, and an additional peak assigned to C-N appeared at ~399.8 eV[13]. In the C *1 s* spectra of the three electrolytes, peaks of the commonly observed species of C–C (~284.8 eV), C=O (~288.7 eV), and C–O (~286.1 eV)[13,28] were observed. However, PVFH-PVCA showed a higher intensity of the C=O peak and an additional signal of C-F (290.8 eV)[47]. The presence of C-F (~688.4 eV) was further confirmed from the F *1 s* spectrum[48], in which two peaks appeared at ~685 and ~686.4 eV, corresponding to LiF and Li$_x$PO$_y$F$_z$[43], respectively. The SEI formed in the system with PVFH-PVCA had a higher content of F and N (Supplementary Fig. S22). The formation of fluorine-rich and nitrogen-rich interphases contributes to enhanced interfacial kinetics and stability[49,50].

We further investigated the nanostructural features of the SEI using cryo-transmission electron microscopy (Cryo-TEM). In general, compared with the Li dendrites in CLE cells and dead Li in PVLE cells, PVFH-PVCA could induce massive deposition of Li (Supplementary Fig. S23); consequently, a continuous and uniform SEI was formed, as shown in Fig. 5d. When the magnification was increased to the atomic scale, a dual-layered SEI with an inorganic inner phase and an amorphous outer layer in the polymer system was identified (Fig. 5e). The inner inorganic-rich layer comprised a small amount of Li$_3$N and a large amount of LiF, consistent with the XPS results. Cryo-STEM allowed

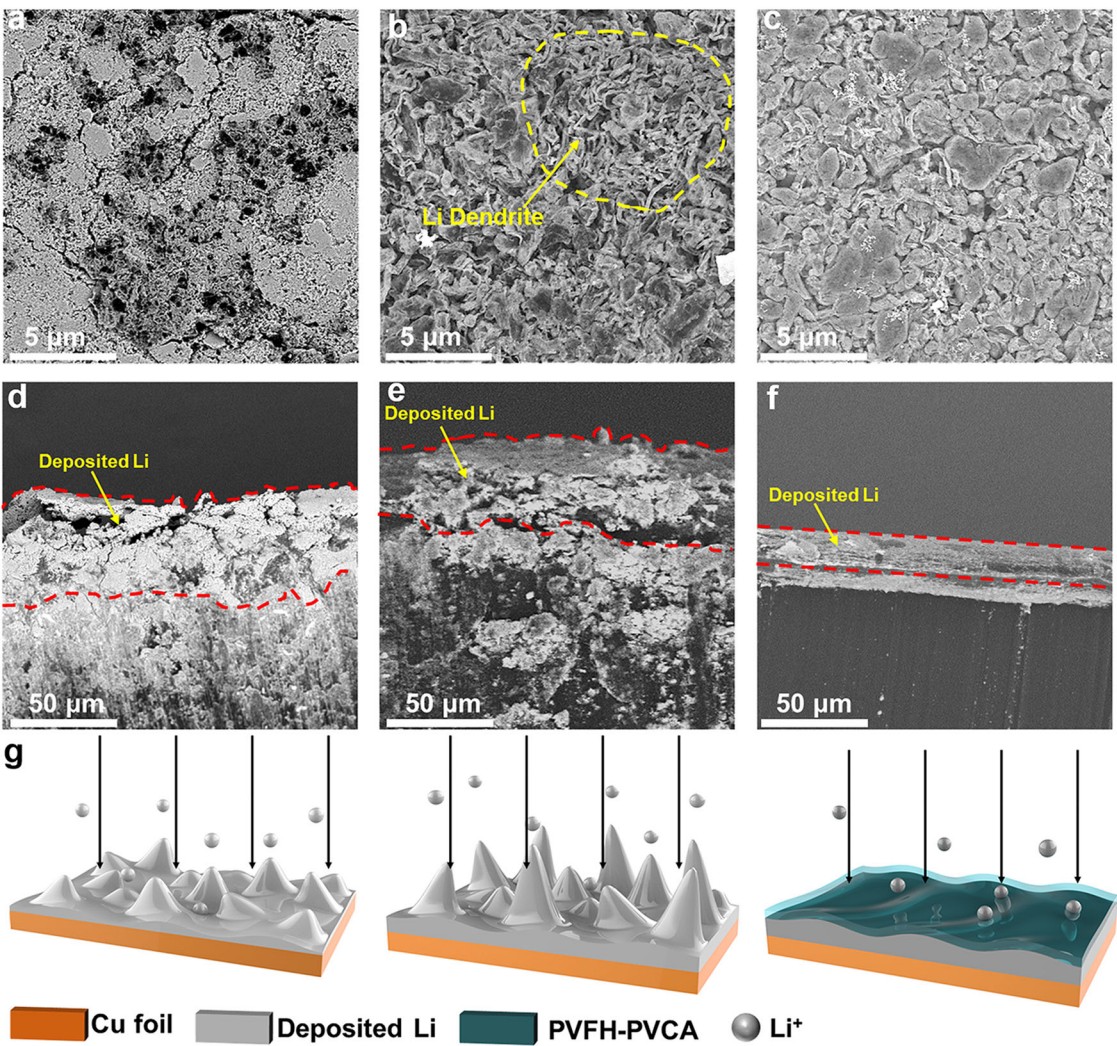

**Fig. 4 | SEM study of the interfacial Li-deposition behavior. a–f** Top-view and cross-sectional SEM images of Li deposits obtained by plating (capacity: 4 mAh cm⁻²) Li on a Cu substrate at 8 mA cm⁻² in Li||Cu cells with **a** and **d**, PVLE, **b** and **e**, CLE, and **c** and **f**, PVFH-PVCA. **g** Schematic illustration of the Li-deposition behavior in cells with different electrolytes.

further visualization of a homogeneous SEI with a thickness of ~12 µm for the PVFH-PVCA cell (Fig. 5f). In contrast, the SEI formed in the CLE cell was thicker and inhomogeneous (Supplementary Fig. S24a and c). Cryo-electron energy loss spectroscopy (Cryo-EELS) revealed that the SEI was rich in N (Fig. 5g and Supplementary Fig. S25) and that F and N were uniformly distributed in the SEI, which facilitates improved interfacial dynamics and stability, unlike in the case of the SEI formed by the CLE cell without N (Supplementary Fig. S24b and S25).

## Application of the PVFH-PVCA electrolyte in high-energy-density LMBs

To demonstrate the applicability of PVFH-PVCA as a prospective polymer electrolyte, its electrochemical characteristics were studied in LMBs fabricated with various cathodes, including $Li_{1.2}Mn_{0.56}Ni_{0.16}Co_{0.08}O_2$ (LMNCO), LCO, LFP, and NC95. Before evaluating the battery performance, electrochemical floating experiments were conducted on the cells with PVFH-PVCA to accurately determine their actual operational electrochemical windows (Supplementary Fig. S26)[14,51]. For the PVFH-PVCA, the measured leakage current was less than 20 µA up to a voltage of 4.9 V. Thus, the excellent oxidative stability of PVFH-PVCA allows for stable operation (cycling and rate capability) with LMNCO cathodes at a high voltage of 4.8 V (Fig. 6c, d). The cyclic voltammogram revealed that the LMNCO|PVFH-PVCA|Li battery exhibits superior redox reaction

reversibility and kinetics compared with those of the systems based on CLE and PVLE (Supplementary Fig. S27), suggesting that the PVFH-PVCA has excellent interfacial stability with both the LMNCO cathode and Li-metal anode. The cycling performance revealed that the LMNCO|PVFH-PVCA|Li battery has a capacity retention of 84.8% (initial capacity: 214.1 mA h g⁻¹) after 400 cycles at 2.1–4.6 V and 1 C (Fig. 6a). Evidently, it outperforms other excellent polymer electrolytes developed in earlier works. In clear contrast, the LMNCO|CLE|Li and LMNCO|PVLE|Li batteries exhibited lower capacity retentions of only 64.0 and 33.9%, respectively, after cycling under the same conditions. In addition, the pristine LMNCO|CLE|Li battery presented a charge-transfer resistance ($R_{ct}$) of ~121 Ω, which increased substantially to ~887 Ω after 300 cycles, whereas the $R_{ct}$ of the LMNCO|PVFH-PVCA|Li battery increased from ~126 to ~248 Ω after 300 cycles (Supplementary Fig. S28). Figure 6b shows that the charge/discharge curves of LMNCO|PVFH-PVCA|Li are flat and the voltage polarization remains small after cycling. In addition, the capacity retentions after 340 cycles at 2.1–4.8 V and 1 C were as high as 78.8% (initial capacity: 238.4 mA h g⁻¹; Fig. 6c), and the voltage decay was also significantly improved (Supplementary Fig. S29). In contrast, the LMNCO|CLE|Li and LMNCO|PVLE|Li batteries were damaged after only 207 and 100 cycles. The specific capacities of the LMNCO|PVFH-PVCA|Li batteries at 0.1, 0.5, 1.0, 2.0, 4.0, 6.0, and 8.0 C rates are 287.2, 251.1, 215.9, 177.8, 149.1, 126.8, and 109.2 mAh g⁻¹, respectively, much

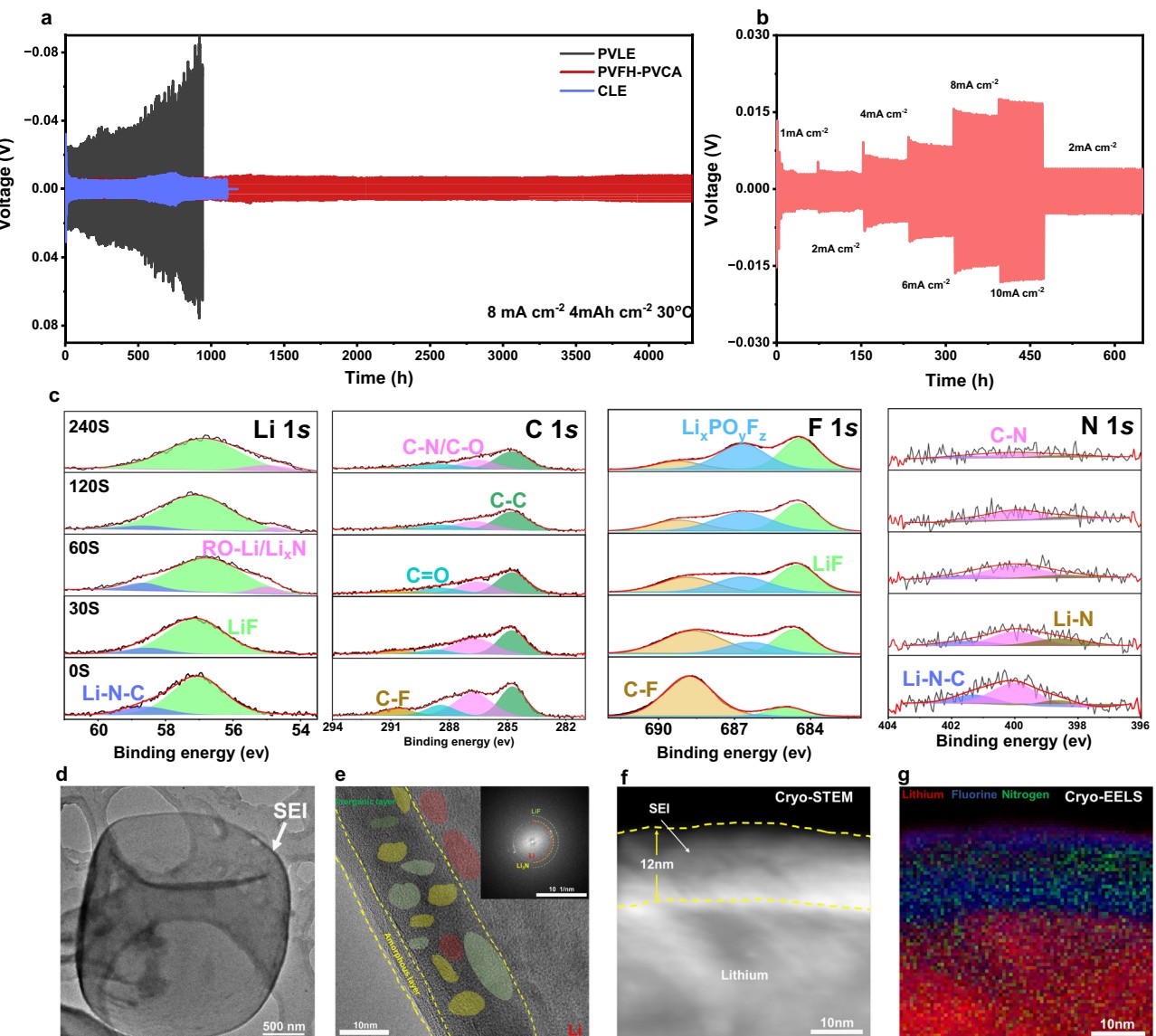

**Fig. 5 | Characterization of Li ‖ Li symmetric cells. a** Galvanostatic voltage profiles of Li‖Li symmetric cells with different electrolytes at 8.0 mA cm⁻² and 4.0 mAh cm⁻². **b** Voltage versus time curves at current densities of 1.0 to 10 mA cm⁻² for the PVFH-PVCA cell. **c** XPS depth profiles of Li 1 s, C 1 s, F 1 s, and N 1 s in Li-metal anodes cycled in Li‖Li coin cells with the designed polymer electrolyte. **d**, **e** Cryo-TEM images of Li deposited in the cell with the PVFH-PVCA at different scales. Inset in **e** shows the fast Fourier transform pattern of the inner SEI: LiF (green circle), Li (red circle), and Li₃N (yellow circle). **f** Cryo-STEM image of Li deposited in the cell with PVFH-PVCA and **g** the corresponding Cryo-EELS image.

higher than those of the LMNCO|CLE|Li battery (Fig. 6d). More importantly, the cycling and rate performances of the LMNCO‖Li using the PVFH-PVCA electrolyte were significantly improved compared with those of the batteries with other electrolytes (Supplementary Fig. S30-32).

To understand how PVFH-PVCA stabilizes the LMNCO cathode, the LMNCO cathode (removed from the LMNCO‖Li button cell after 50 cycles at 1 C and 2.1–4.8 V) was sampled and studied in situ. The TEM images in Fig. 6e reveal that after cycling in PVFH-PVCA, a thin and uniform CEI layer (~8 nm) was formed on the surface of the LMNCO cathode, which facilitated the suppression or delay of extensive intergranular cracking of LRMO particles[52] and ensured stable cycling of the cell. In contrast, a thicker and non-uniform CEI layer was observed on the surface of the LMNCO cathode cycled in CLE and PVLE, as confirmed by the TEM images in Fig. 6f, g. To further characterize the chemical states and components of the CEIs, XPS of the surfaces of the cathodes was conducted after 50 cycles (Supplementary Fig. S33). The peaks of $CO_3^{2-}/O–C=O$

(~ 292.1 eV) and −COOR (287.5 eV) were clearly observed in the XPS profiles, indicating undesired electrolyte decomposition on the surfaces of CLE|LMNCO and PVLE|LMNCO. The O 1 s XPS results revealed that the lattice oxygen (529.5 eV) peak of LMNCO at the PVFH-PVCA|LMNCO interface was stronger, indicating that a thinner CEI was formed on the LMNCO surface, which is consistent with the results in Fig. 6e. Further, strong C-N signals were observed for the PVFH-PVCA/LMNCO interface, indicative of Li-N-C rich inorganic components[53].

Although the LMNCO|PVFH-PVCA|Li coin cells exhibit an outstanding electrochemical performance, high loading is desirable to achieve a comparable area capacity to that of commercialized cells. Figure 6h shows a highly loaded LMNCO|PVFH-PVCA|Li pouch cell cycled at room temperature with an initial surface capacity of ~2.3 mAh cm⁻² (10 mg cm⁻²) at 1 C and an area capacity retention rate of 85.2% after 160 cycles. In addition, at higher cut-off voltages, the LMNCO|PVFH-PVCA|Li pouch cell could be operated stably for more than 120 cycles

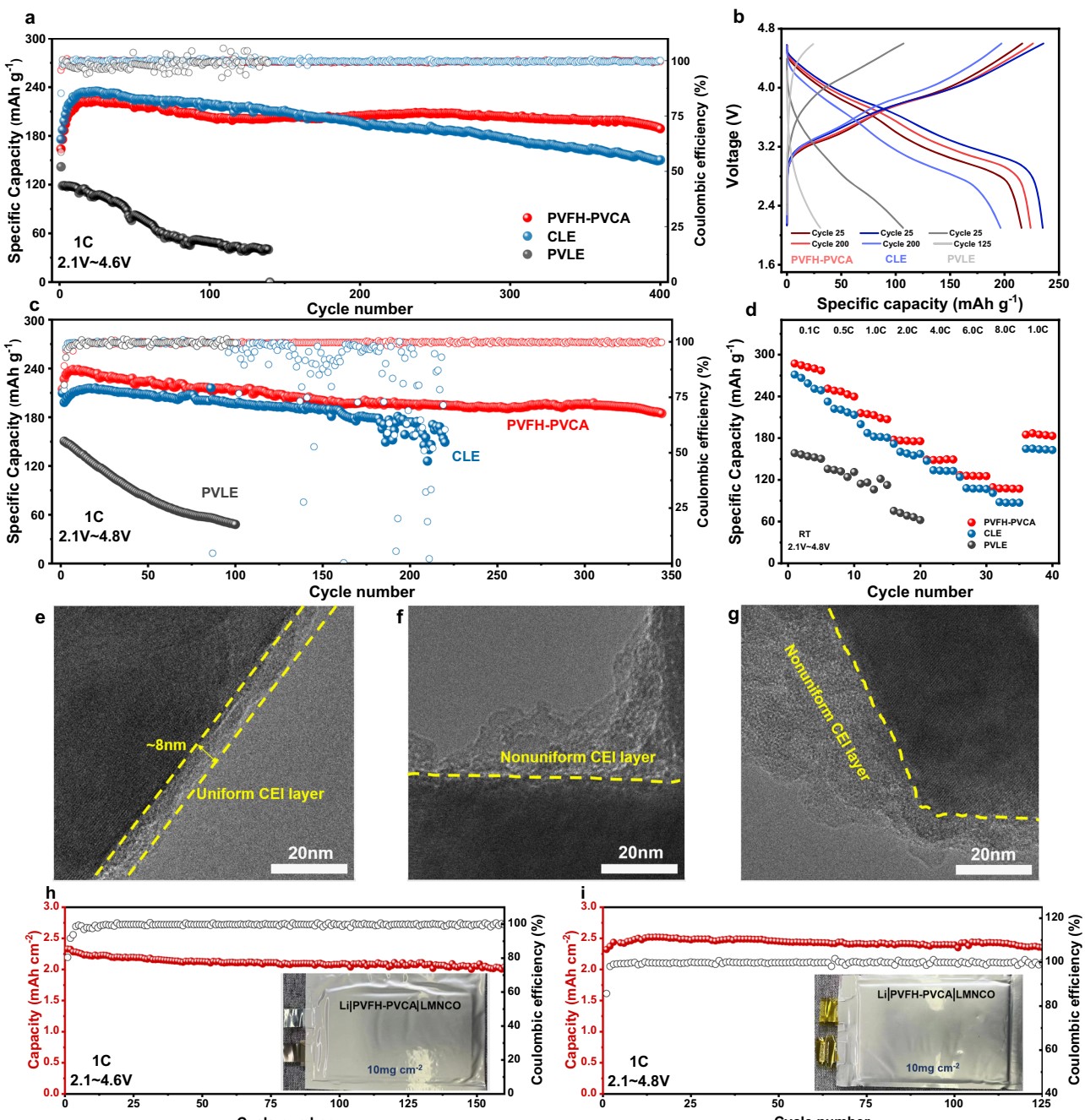

**Fig. 6 | Electrochemical performance of the lithium-metal batteries with the Li$_{1.2}$Mn$_{0.56}$Ni$_{0.16}$Co$_{0.08}$O$_2$ (LMNCO) cathode and different electrolytes. a** Cycling performance of the Li||LMNCO cells with CLE, PVLE, and PVFH-PVCA at 1 C between 2.1 and 4.6 V. **b**, Charge/discharge curves at the 25th and 125th cycles for the Li||LMNCO cells with CLE, PVLE, and PVFH-PVCA at 1 C. **c** Cycling performance of the Li||LMNCO cells with CLE, PVLE, and PVFH-PVCA at 1 C between 2.1 and 4.8 V.

**d**, Rate capability of the Li||LMNCO cells with CLE, PVLE, and PVFH-PVCA from 0.1 to 8.0 C. TEM images of the LMNCO particles in the cells using PVFH-PVCA (**e**), CLE (**f**), and PVLE (**g**). **h** and **i**, Long-term cycling performance of the exemplary Li|| LMNCO soft pack cells employing a single layer anode at 2.1–4.6 V (**h**) and 2.1–4.8 V (**i**).

To further verify the electrochemical performance of the PVFH-PVCA electrolyte, LFP|PVFH-PVCA|Li, NC95|PVFH-PVCA|Li, and LCO|PVFH-PVCA|Li full coin cells were assembled and evaluated (Fig. 7). High-voltage LCO|PVFH-PVCA|Li cells (1 C, 3.0–4.5 V) with an average discharge voltage of ~4 V exhibited a discharge capacity retention of 89.2% after 1500 cycles (Fig. 7a–c). For the LFP|PVFH-PVCA|Li full cell, steady cycling stability of 86.4% was obtained after 1940 cycles (0.5 C, 2.6–3.8 V; Fig. 7d–f). In addition, the cycling stability of the high-voltage NC95 cathode was also improved, and the NC95|PVFH-PVCA|Li cell showed 78.7% discharge capacity retention (1 C, 2.7–4.3 V) after

200 cycles (Fig. 7g–i). These results confirm the universality and compatibility of the PVFH-PVCA electrolyte with various electrode materials.

In summary, we successfully developed an entanglement association polymer electrolyte (PVFH-PVCA) with superior mechanical properties, high ionic conductivity, excellent interface compatibility, electrochemical stability, and high Li-ion transference number by incorporating the PVCA terpolymer. The inherent low oxidation potential and the lowest reduction potential of PVCA enable it to preferentially oxidize and form a thin, uniform, and robust CEI on the

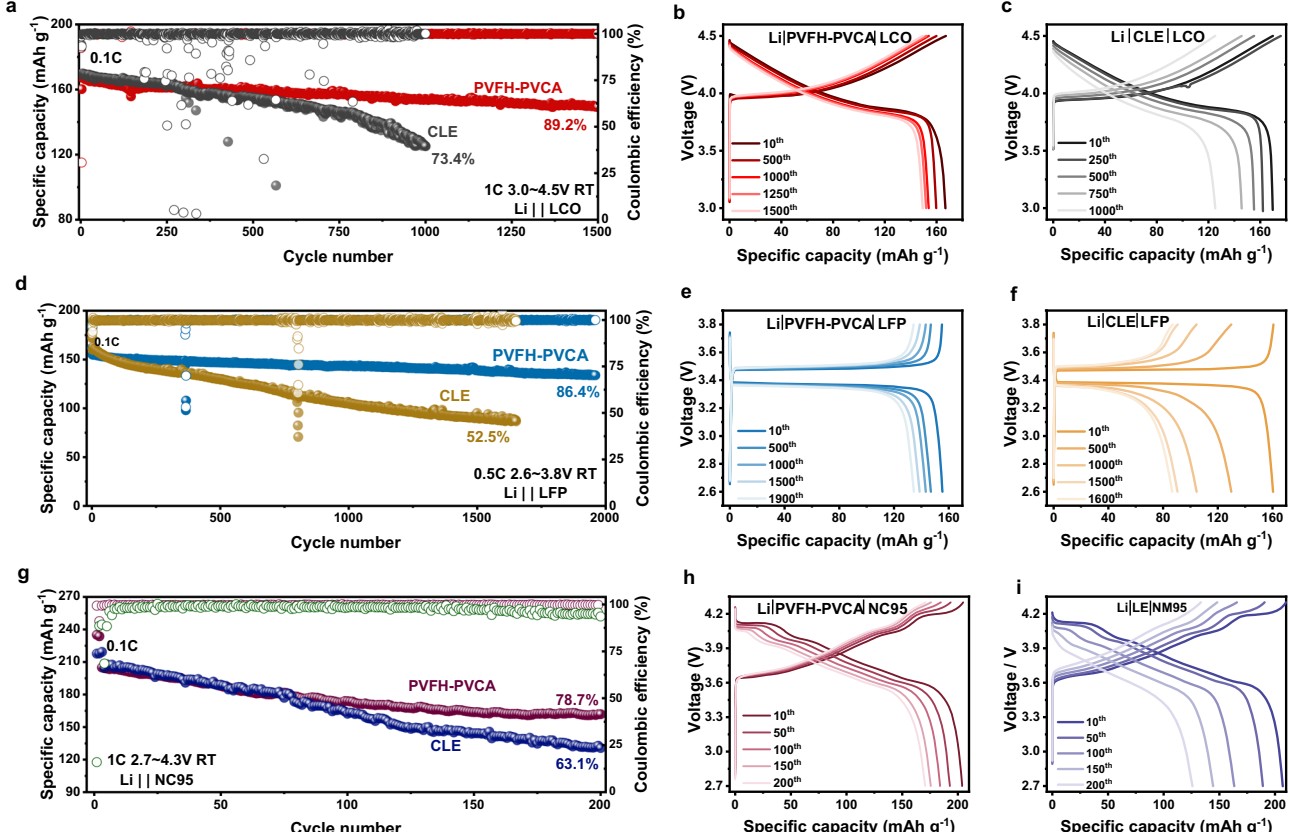

**Fig. 7 | Electrochemical performance of the lithium-metal batteries with different cathodes and different electrolytes (CR2032 coin cells, 30°C).**
**a** Comparison of the cycling performances of LiCoO$_2$ (LCO)|PVFH-PVCA|Li and LCO|CLE|Li full cells at 1 C. **b**, **c** Comparison of the charge/discharge voltage profiles of **b**, LCO|PVFH-PVCA|Li and **c** LCO|CLE|Li full cells. **d** Comparison of the cycling performance of LiFePO$_4$ (LFP)|PVFH-PVCA|Li and LFP|CLE|Li full cells at

0.5 C. Comparison of the charge/discharge voltage profiles of **e**, LFP|PVFH-PVCA|Li and **f**, LFP|CLE|Li full cells. **g** Comparison of the cycling performances of LiNi$_{0.95}$Co$_{0.05}$O$_2$ (NC95)|PVFH-PVCA|Li and NC95|CLE|Li full cells at 1 C. Comparison of the charge/discharge voltage profiles of **h**, NC95 | PVFH-PVCA|Li and **i**, NC95|CLE|Li full cells.

LMNCO cathode surface; this effectively suppresses side reactions with the electrolyte. Additionally, the preferential participation of PVCA in SEI formation at the Li anode leads to a dual-layered SEI enriched in N and F, which differs significantly from the conventional SEI structure. Thus, it inhibits the formation of "dead" Li and promotes dendrite-free Li deposition, thereby enabling symmetrical Li//Li batteries to achieve a remarkable cycling stability for up to 4500 h at a high current density of 8 mA cm$^{-2}$. Furthermore, the Li| PVFH-PVCA| LMNCO batteries exhibit excellent cycling stability for more than 340 cycles at a cut-off voltage of 4.8 V. Further, high-loading Li||LMNCO pouch batteries also exhibit outstanding cycling and a high surface capacity. The PVFH-PVCA hybrid design has significant merits in terms of compatibility and can be extended to other conventional cathode materials to realize excellent cycle stability. This study provides a new avenue for high-energy-density Li–LRMO batteries.

## Methods
### Materials and PVCA synthesis
All reagents were purchased from commercial suppliers and used without further purification, unless otherwise stated. All procedures involving air- and/or water-sensitive compounds were carried out using standard Schlenk and vacuum line techniques or glove box techniques under an argon atmosphere (H$_2$O and O$_2$ < 0.1 ppm). AN (98%, Aladdin), MA (99%, Sigma-Aldrich), and VC (99%, Aladdin) were mixed at a molar ratio of 9:1:9, and then 0.15 wt% azobisisobutyronitrile (AIBN) was dissolved in the mixture. The resulting solution was maintained under an argon atmosphere at 60 °C for 15 h. The resultant

solid was dissolved in anhydrous Dimethyl sulfoxide (DMSO) and washed with methylbenzene for removing any impurities, such as residual monomers and the initiator, to obtain pure PVCA.

### Preparation of the PVFH-PVCA electrolyte
First, 0.5 g of PVFH (average Mw: ~400,000, average Mn ~130,000; 98%, Adamas) was added to 6 mL of DMSO and stirred at 60 °C for 6 h. Separately, PVCA was dissolved in DMSO to a concentration of 10 wt% at 70 °C for 2 h. Then, the two solutions were mixed to obtain a series of PVFH/PVCA mixtures with different PVCA concentrations and stirred at 60 °C for 12 h. The PVFH/PVCA mixtures were then cast on polytetrafluoroethylene plates to form PVFH-PVCA membranes (~40 μm thickness). After vacuum-drying at 60 °C for 12 h, the PVFH-PVCA membrane was transferred to an Ar-filled glovebox and immersed in a certain amount of LB-372 electrolyte (1.0 M LiPF$_6$ and 0.02 M LiDFOB in FEC:HFE:FEMC = 2:2:6 vol%) for 24 h to obtain the PVFH-PVCA electrolyte. Notably, due to the preparation temperature(60 °C) of the PVFH-PVCA membrane being much lower than he individual decomposition temperatures of PVFH and PVCA (å 300 °C), and PVFH is physically mixed with PVCA to obtained the entangled structure, the exact ratio of PVFH to PVCA with various percentage is the same as the feeding ratio of PVFH and PVCA. For comparison, the PVFH electrolyte was also prepared in the same way without adding PVCA. Before assembling the batteries, the excess LE on the surface was removed with filter paper.

Electrode preparation. LMNCO, NC95, LCO, and LFP cathodes were prepared using a slurry-casting technique. The active material,

Super P carbon, and polyvinylidene fluoride were dissolved in N-methyl-2-pyrrolidone at a weight ratio of 8:1:1 to prepare a slurry, and the slurry was coated on a current collector of aluminum foil. The cathodes were dried in a vacuum oven at 110 °C for 24 h. The active loading densities of the LMNCO, NC95, LCO, and LFP cathodes were 2.5–4.5, ~3.0, ~3.2, and ~3.0 mg cm$^{-2}$, respectively.

## Material characterization

$^{1}$H-NMR and $^{13}$C-NMR spectra were performed on a Bruker Avance NEO 600 NMR spectrometer. FTIR spectra were acquired on a Bruker Vertex 70 v spectrometer. Raman spectroscopy was performed on a Micro-laser confocal Raman spectrometer (Horiba LabRAM HR800, France) using a 532-nm-wavelength laser. The GPC system is a Shimadzu LC-20AD pump system consisting of an auto injector, a MZ-Gel SDplus 10.0 μm guard column (50 × 8.0 mm, 10$^{2}$ Å) followed by two PLgel 5 μm MIXED-D columns (300 × 7.5 mm), and a Shimadzu RID-10A refractive index detector[54]. The morphology and elemental distribution were characterised using a field emission scanning electron microscope (SU-70 FESEM) coupled with an energy dispersive X-ray spectrometer. The stress-strain curves of the polymer electrolytes were obtained using a tensile tester at a tensile speed of 0.2 cm/min. XPS was performed using monochromatic Al Kα radiation on a high resolution Kratos AXIS 165 X-ray photoelectron spectrometer. Depth profiling of the Li electrode was carried out using argon ion sputtering. The cycled Li and LMNCO electrodes were adequately washed with dimethoxyethane and dried in a glove box. TEM grids were loaded on a cryogenic vacuum transfer holder (Fischione 2550) in an ar-filled glove box and transferred to the TEM chamber without exposure to air. The samples were kept at a low temperature (−180 °C) throughout the experiment. AFM (Bruker-Fastscan) was performed in a dry environment (XY maximal scanning range: 90 μm × 90 μm, Z ≤ 3 μm).

## Electrochemical measurements

The electrolyte film was sandwiched between two stainless steel (SS) electrodes with a diameter of 16 mm, and a coin cell (CR2032) of SS| electrolyte|SS was assembled to determine the ionic conductivity. Electrochemical impedance spectroscopy (EIS) was performed using a CHI 660E electrochemical workstation (Shanghai Chenhua Instrument Co., Ltd.) at a temperature range of 25–80 °C; the frequency range was from 0.1 Hz to 1 MHz with an amplitude of 10 mV. The ionic conductivities were calculated using the following equation[6]:

$$\sigma = L/(S^{*}R) \tag{1}$$

where L (cm) is the thickness of the PVFH-PVCA membrane (~45 μm), R (Ω) is the bulk resistance of the electrolyte, and S (cm$^{2}$) is the contact areas of the electrode and electrolyte. The electrochemical stability window of the solid-state polymer electrolyte was determined by LSV using a Li/electrolyte/SS cell in the range of 2.5-7 V at a scan rate of 1 mV/s. The $t_{Li+}$ was determined by the CA tests were performed on lithium symmetric (Li| electrolyte |Li) cells at a potential of 10 mV until the current reached steady state. EIS measurements were performed before and after the polarisation scan. The Coulombic efficiency of Li plating and stripping cycles was investigated using a Li||Cu half-cell structure. In each cycle, 0.5 mA h cm$^{-2}$ of Li was plated on the Cu electrode at a current density of 0.5 mA cm$^{-2}$ and then stripped until the potential reached 1.5 V vs. Li/Li$^{+}$. Galvanostatic charge/discharge tests of symmetric Li||Li cells, Li||LMNCO, Li||LCO, Li||NC95, and Li||LFP cells were performed on a NEWAIE battery tester. For the cells with the LE, the LE amount was controlled at 60 μL in each cell. The current densities of the full cells of Li||LMNCO, Li||LCO, Li||NC95, and Li||LFP are defined as 1 C = 200, 180, 200, and 170 mA g$^{-1}$, respectively. All tests were performed at room temperature, unless specified otherwise. The deposited Li-metal electrodes and cycled Li-metal anode were washed with methoxymethane in a glove box and transferred to the instrument

in a sealed jar to prevent oxidation. Single-layer Li||LMNCO pouch cells were assembled with the designed PVFH-PVCA electrolyte in an Ar glovebox (H$_2$O < 0.1 ppm, O$_2$ < 0.1 ppm) using a single-side-coated LMNCO-based positive electrode (6 × 10 cm$^2$, 10 mg cm$^{-2}$, 2.52 mAh cm$^{-2}$, and 2.1–4.8 V) and single-layer Li metal anode (6.5 × 11 cm$^2$ area; 25 μm thickness).

## Density functional theory (DFT) calculations

DFT calculations were performed with the ORCA code[55,56]. The geometrical structure of the system was optimized at the B3LYP/6-31G(d,p) level of theory. Molecular structures were visualized using the VMD package[57] under the assistance of the Multiwfn program[58]. The electrostatic potential involved in the analyses was evaluated using Multiwfn based on the highly effective algorithm presented in ref. [59] For interaction energy calculation, Molclus program (Tian Lu, Molclus program, Version 1.9.9.9, http://www.keinsci.com/research/molclus.html (accessed: Apr 18, 2023)) was deployed to assist with the configuration search; then, several thermodynamically favored configurations were selected from 500 random configurations for every two-monomer complex and Li-PVCA complex to study their interaction energy with BSSE correction at the B3LYP D3 ma-def2-TZVP(-f) level of theory.

## Data availability

The datasets generated and/or analysed during the current study are available from the corresponding authors on request. The Source data generated in this study are provided in the Source Data file. Source data are provided with this paper.

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

## Acknowledgements
This work was financially supported by the National Key R & D Program of China under 2022YFB2502100 and the National Natural Science Foundation of China (No. 53130202). All support for this research is gratefully acknowledged.

## Author contributions
D.-G. X. and H.C.W. conceived the concept and D.-G.X. supervised the project. Y.L.Y. performed LMNCO cathode materials. C.G. completed the computational work. H.C.W. conceived and designed the experiments and wrote the paper. H.C.W. performed the experiment. D.-G. X., H.C.W. and J.S. carried out the data analysis and discussed. Y.X.Z., T.H.Y., T.C., W.K.X. and K.Z., discussed the results and data analysis. D.-G. X. revised the manuscript. Q. F and X.-F.W. conducted the Cryo-STEM test. H.C.W., Y.L.Y., C.G. and T.C. contributed equally to this work. All the authors participated in the discussion of the study.

## Competing interests
The authors declare no competing interests.
