## [Peer Review File · Nature Communications]

An entanglement association polymer electrolyte for Li-metal batteriesREVIEWER COMMENTS

Reviewer #1 (Remarks to the Author):

This manuscript reported the development of an entanglement association polymer electrolyte (PVFH-PVCA) by incorporating the PVCA terpolymer. Overall, the performance is great with coin cell batteries exhibiting cycling stability for more than 300 cycles at a high cut-off voltage of 4.8 V. The high-loading pouch batteries also exhibit good cycling and a high surface capacity.

However, the structural characterization is questionable and more experiments should be further performed:

1. The proton NMR spectrum is not sufficient to conclude the successful preparation of PVCA, as well as the limited range from 1.5-5.5ppm; the authors should compare the spectra with precursor MA, VC, AN, to confirm the reactivity and amount of precursor left in the product.
2. The proton on polymerized MA is not assigned in Figure S3.
3. Carbon NMR spectra should be also provided, which is another strong evidence to ensure the conversion of C=C to C-C during the polymerization.
4. Molecular weight from GPC should be provided.
5. Is there any method can confirm the exact ratio of PVFH to PVCA with various percentage?

Reviewer #2 (Remarks to the Author):

This work reports the design of new gel electrolytes for lithium metal batteries. The authors synthesize acrylonitrile (AN), maleic anhydride (MA), and vinylene carbonate (VC) copolymer, named PVCA. Blends of poly(vinylidene fluoride-co-hexafluoropropylene) (PVFH), PVCA, and liquid electrolytes were used to prepare gel electrolytes. The authors showed that by introducing PVCA, the electrochemical performance of PVFH gel electrolytes was improved. The authors attributed this observation to the enhanced entanglement between PVFH and PVCA. This is an interesting gel electrolyte system. However, some characteristics and mechanistic understanding of the materials remain unclear, detailed as follows.

1. Some of the fundamental characteristics of the polymer are unclear. For example, PVCA is a copolymer. What are the relative contents of the AN, MA, and VC? What is the molecular weight of PVCA and PVFH?
2. The authors compared PVFH gel and PVFH-PVCA gel; it is however unclear if the sol content of these two gels is the same. If not, the differences in physical and electrochemical properties could be due to the different sol contents.
3. The XRD characterization is quite confusing. The authors mentioned, "For mixtures of PVCA and PVFH, the diffraction peak intensity gradually increased with the increase in the PVCA content at first and then gradually decreased (Supplementary Fig. S5a). This result indicates a strong interaction between PVCA and the PVFH matrix.". Which peak did the authors refer to? It seems the intensity of most peaks decreases. How does the XRD pattern tell the interaction between these two polymers? Assigning the diffraction peaks first would be helpful, and then referring to specific diffraction peaks in the discussion. How was the XRD sample prepared? The sample preparation procedure might affect the results.
4. The modulus data from tensile and AFM are off by nearly three orders of magnitudes.
5. Figure 3a shows that the oxidation potential of PVFH-PVCA is also lower than 5 V, as there is a small step before 5V.
6. Figure 5a, are the samples mislabeled? The figure shows CLE has the longest cycling life.

Response to reviewers

We gratefully thank all the reviewers for their time spent making their constructive remarks and useful suggestions, which can enable us to make improvements and significantly raise the quality of the manuscript. Each suggested revision and comment, brought forward by the reviewers was accurately incorporated and considered. Below the comments of the reviewers are addressed point-by-point and the revisions are indicated.

Response to reviewer's comments

Reviewer #1 (Remarks to the Author):

This manuscript reported the development of an entanglement association polymer electrolyte (PVFH-PVCA) by incorporating the PVCA terpolymer. Overall, the performance is great with coin cell batteries exhibiting cycling stability for more than 300 cycles at a high cut-off voltage of 4.8 V. The high-loading pouch batteries also exhibit good cycling and a high surface capacity. However, the structural characterization is questionable and more experiments should be further performed:

We appreciate the reviewer's positive comments

Comment 1: The proton NMR spectrum is not sufficient to conclude the successful preparation of PVCA, as well as the limited range from 1.5-5.5ppm; the authors should compare the spectra with precursor MA, VC, AN, to confirm the reactivity and amount of precursor left in the product.

Response: Thanks for reviewer's professional suggestion. We have expanded the NMR spectrum range from -0.1 to 12.5 ppm in the revised manuscript, as shown in Figure R1d. The ¹H-NMR spectrum of PVCA was analyzed and the peaks in the spectrum were attributed. We found that the distinct peaks appeared around the low field (2-3.5 ppm), which can indicate the appearance of hydrogen atoms connected by C-C single bonds. Additionally, the peaks in the proton NMR spectrum were attributed and integrated. From the integration results we can find that the ratio of the number of substrates involved in the reaction in PVAC is MA:VC:AN=1:3:3. We have made corresponding changes in the manuscript and marked them in yellow. Furthermore, by analyzing the ¹³C-NMR spectra of the reactants MA, VC, AN, and PVAC (Figure. R2), it can be observed that the peaks of C=C double bonds (100-140 ppm) can be observed in the reactants MA, VC, and AN. The peak intensity of C=C double clicking in the product PVAC is significantly reduced, and there is a clear C-C single bond peak (20-40ppm) in the low field, confirming that the substrate did undergo polymerization reaction. All of them suggest that the polymerization reaction did occur.

Figure R1. $^1\text{H-NMR}$ spectrum of MA(a), AN(b), VC(c) and PVCA(d).

Changes made in the revised manuscript:

We replaced the pictures **Supplementary Fig. 3** with modified versions of **Supplementary Fig. 3a** and inserted the discussion of $^1\text{H-NMR}$ into the first paragraph in **Synthesis and characterization of PVCA** section, as follows: “Furthermore, the $^1\text{H-NMR}$ spectrum of PVCA was analyzed and the peaks in the spectrum were attributed (Supplementary Fig. 3a). It can be found that the distinct peaks appeared around the low field (2-3.5 ppm), which can indicate the appearance of hydrogen atoms connected by C-C single bonds. Additionally, the integration results of the peaks show that the ratio of the number of substrates involved in the reaction in PVAC is MA:VC:AN=1:3:3. By analyzing the $^{13}\text{C-NMR}$ spectra of the reactants MA, VC, AN, and PVAC (Supplementary Fig. 3b-e), it can be observed that the peaks of C=C double bonds (100-140 ppm) can be observed in the reactants MA, VC, and AN. The peak intensity of C=C double clicking in the product PVAC is significantly reduced, and there is a clear C-C single bond peak (20-40ppm) in the low field, confirming that the substrate did undergo polymerization reaction.”

Supplementary Fig. 3(a) ^1H NMR spectrum of the PVCA.

Comment 2. The proton on polymerized MA is not assigned in Figure S3.

Response: We sincerely thank the reviewer for pointing out this critical detail. We have modified the **Supplementary Fig. 3a**.

Changes made in the revised manuscript:

We labeled the proton of MA after polymerization as “d” in the illustration of **Supplementary Fig. 3a**.

Comment 3. Carbon NMR spectra should be also provided, which is another strong evidence to ensure the conversion of C=C to C-C during the polymerization.

Response: Thanks for reviewer’s professional advice. In the revised Support information, we have added the carbon NMR spectra for various monomers and PVCA. By analyzing the ^{13}C -NMR spectra of the reactants MA, VC, AN, and PVAC (Figure. R2), it can be observed that the peaks of C=C double bonds (100-140 ppm) can be observed in the reactants MA, VC, and AN. The peak intensity of C=C double bonds in the product PVAC is significantly reduced, and there is a clear C-C single bond peak (20-40ppm) in the low field, confirming that the substrate did undergo polymerization reaction. We have made corresponding changes to the revised support information and marked it in yellow.

Figure R2. ^{13}C -NMR spectrum of some compounds mentioned in this paper. (a) ^{13}C -NMR spectrum of MA. (b) ^{13}C -NMR spectrum of AN. (c) ^{13}C -NMR spectrum of VC. (d) ^{13}C -NMR spectrum of PVCA.

Changes made in the revised manuscript:

We have included the ^{13}C -NMR data in **Supplementary Fig. 3**, labeled as **Supplementary Fig. 3b-e**. Furthermore, we have inserted the discussion of ^{13}C -NMR into the first paragraph in **Synthesis and characterization of PVCA** section, as follows: “By analyzing the ^{13}C -NMR spectra of the reactants MA, VC, AN, and PVAC (Supplementary Fig. 3b-e), it can be observed that the peaks of C=C double bonds (100-140 ppm) can be observed in the reactants MA, VC, and AN. The peak intensity of C=C double clicking in the product PVAC is significantly reduced, and there is a clear C-C single bond peak (20-40ppm) in the low field, confirming that the substrate did undergo polymerization reaction.”

Supplementary Fig. 3 | ^{13}C -NMR spectrum of the MA (b). ^{13}C -NMR spectrum of AN (c). ^{13}C -NMR spectrum of VC (d). ^{13}C -NMR spectrum of PVCA (e).

Comment 4. Molecular weight from GPC should be provided.

Response: We sincerely thank the reviewer for the suggestions. In the revised manuscript, we measured the Molecular weight of PVCA using Gel Permeation Chromatography. As shown in Fig R3, the Molecular weight of PVCA is 91100. We have made corresponding changes in the manuscript and marked them in yellow.

Figure R3. GPC curves of PVCA.

Changes made in the revised manuscript:

We have included the GPC data in **Supplementary Fig. 4a**. Furthermore, we have inserted the

discussion of GPC into the first paragraph in **Synthesis and characterization of PVCA** section, as follows: "As shown in Supplementary Fig. 4a, the molecular weight of PVCA was determined to be 91,100 using Gel Permeation Chromatography (GPC)."

Supplementary Fig. 4(a) GPC curves of PVCA.

Comment 5. Is there any method can confirm the exact ratio of PVFH to PVCA with various percentage?

Response: Thanks for reviewer's question. In the initial manuscript, we did not express this issue clearly. In the revised manuscript, we have added a new description. In present work, thermogravimetric analysis demonstrates that the individual decomposition temperatures of PVFH and PVCA are both above 300°C. The preparation process of PVFH-PVCA membrane is less than 80°C and PVFH is physically mixed with PVCA to obtained the entangled structure. As a result, there is no material loss due to volatilization during the film-forming process, and no phase separation is observed after film formation, indicating uniform film formation at the feeding ratio of the two components. We have made corresponding changes in the manuscript and marked them in yellow.

Changes made in the revised manuscript: In the revised manuscript, we have added a new description into the **Preparation of the PVFH-PVCA electrolyte** section, as follows: "Notably, due to the preparation temperature(60°C) of the PVFH-PVCA membrane being much lower than he individual decomposition temperatures of PVFH and PVCA (300°C), and PVFH is physically mixed with PVCA to obtained the entangled structure, the exact ratio of PVFH to PVCA with various percentage is the same as the feeding ratio of PVFH and PVCA."

Reviewer #2 (Remarks to the Author):

This work reports the design of new gel electrolytes for lithium metal batteries. The authors synthesize acrylonitrile (AN), maleic anhydride (MA), and vinylene carbonate (VC) copolymer, named PVCA. Blends of poly(vinylidene fluoride-co-hexafluoropropylene) (PVFH), PVCA, and liquid electrolytes were used to prepare gel electrolytes. The authors showed that by introducing PVCA, the electrochemical performance of PVFH gel electrolytes was improved. The authors attributed this observation to the enhanced entanglement between PVFH and PVCA. This is an interesting gel electrolyte system. However, some characteristics and mechanistic understanding of the materials remain unclear, detailed as follows.

Response: We appreciate the reviewer's positive assessment of our designed polymer electrolyte and the constructive professional feedback, which greatly contributes to the improvement of the quality of the manuscript.

Comment 1. Some of the fundamental characteristics of the polymer are unclear. For example, PVCA is a copolymer. What are the relative contents of the AN, MA, and VC? What is the molecular weight of PVCA and PVFH?

Response: Thanks for reviewer's professional comments. The $^1\text{H-NMR}$ spectrum of PVCA (Figure R4a) was analyzed and the peaks in the spectrum were attributed and integrated. From the integration results it can be found that the ratio of the number of substrates involved in the reaction in PVAC is MA:VC:AN=1:3:3. PVFH average Mw: ~400,000, average Mn ~130,000; 98%, Adamas; The molecular weight of PVCA measured by Gel Permeation Chromatography (GPC) was 91100 (Figure R4b). We have made corresponding changes in the revised manuscript and marked them in yellow.

Figure R4. (a) $^1\text{H-NMR}$ spectrum of PVCA. (b) GPC curves of PVCA.

Changes made in the revised manuscript:

We replaced the pictures **Supplementary Fig. 3** with the revised version of **Supplementary Fig. 3a** and included the GPC data in **Supplementary Fig. 4**, labeled as **Supplementary Fig. 4a**. We have inserted the discussion of $^1\text{H-NMR}$ and GPC into the first paragraph in **Synthesis and characterization of PVCA** section, as follows: “Furthermore, the $^1\text{H-NMR}$ spectrum of PVCA was analyzed and the peaks in the spectrum were attributed (Supplementary Fig. 3a). It can be found that the distinct peaks appeared around the low field (2-3.5 ppm), which can indicate the appearance of hydrogen atoms connected by C-C single bonds. Additionally, the integration results of the peaks show that the ratio of the number of substrates involved in the reaction in PVAC is MA:VC:AN=1:3:3. As shown in Supplementary Fig. 4a, the molecular weight of PVCA was determined to be 91,100 using Gel Permeation Chromatography (GPC).”

Supplementary Fig. 3(a) ^1H NMR spectrum of the PVCA.

Supplementary Fig. 4(a) GPC curves of PVCA.

Comment 2. The authors compared PVFH gel and PVFH-PVCA gel; it is however unclear if the sol content of these two gels is the same. If not, the differences in physical and electrochemical properties could be due to the different sol contents.

Response: Thanks for reviewer’s professional question. The sol content tests on PVFH and PVFH-PVCA samples were measured following the procedure described in relevant literature¹. Specifically, dry films of PVFH and PVFH-PVCA were thoroughly dried in a vacuum oven at 80°C for 24 hours, and the mass was recorded as W_0 . Subsequently, the dry film was immersed in an electrolyte for 72 hours, excess liquid on the surface was wiped off, and their masses were recorded as W_i . Finally, the

gel fraction was calculated according to Equation 1:

$$\text{Gel fraction [\%]} = \frac{W_i - W_0}{W_0} \times 100 \quad 1$$

Each membrane was subjected to five sets of parallel experiments, and the average values were taken. Finally, the gel fraction for PVFH was determined to be 11.2%, and for PVFH-PVCA, it was 12.8%. Different swelling behaviors are attributed to the wetting and adsorption of the solution by the membranes, both before and after modification, which is one of the directions for polymer electrolyte modification^{2,3}. From the test data, the differences in swelling between the two are not significant, indicating limited impact on the physical and electrochemical properties. To further eliminate the influence of swelling, we deliberately controlled the liquid absorption of PVFH and PVFH-PVCA at 10% (adding an amount of liquid electrolyte equal to 10% of the dry polymer film mass, ensuring complete wetting and absorption), and a series of electrochemical tests were conducted. As shown in Figure R5a, under the same sol content, the electrochemical stability window for PVFH-PVCA is ~5.1V, which is higher than that of PVFH (~4.6V). The room temperature ionic conductivity of PVFH-PVCA is ~1.64 mS cm⁻¹, higher than PVFH (~0.21 mS cm⁻¹) (Figure R5b). Additionally, the interface impedance of Li//Cu half-cells after standing for 12 hours also shows that the interface impedance of PVFH-PVCA is significantly lower than that of PVFH (Figure R5c). These results further emphasize that the differences in electrochemical properties are primarily attributed to the different polymer electrolytes.

Figure R5. (a) Electrochemical stability window of PVFH-PVCA with 10% LE and PVLE with 10% LE evaluated by linear sweep voltammetry. (b) EIS of SS/electrolytes/SS batteries. To ensure sufficient electrolyte adsorption, we increased the thickness of the polymer membrane, with PVFH-PVCA with 10% LE having a thickness of 144 μm and PVLE with 10% LE having a thickness of 207 μm ; both electrolyte membranes had an $R = 0.8$ cm. (c) Nyquist plots of Li//Cu half batteries at selected cycles in different polymer electrolytes.

Comment 3. The XRD characterization is quite confusing. The authors mentioned, “For mixtures of PVCA and PVFH, the diffraction peak intensity gradually increased with the increase in the PVCA content at first and then gradually decreased (Supplementary Fig. S5a). This result indicates a strong interaction between PVCA and the PVFH matrix.”. Which peak did the authors refer to? It seems the intensity of most peaks decreases. How does the XRD pattern tell the interaction between these two polymers? Assigning the diffraction peaks first would be helpful, and then referring to specific diffraction peaks in the discussion. How was the XRD sample prepared? The sample preparation procedure might affect the results.

Response: Thanks for reviewer' question. Figure R6a was normalized in the revised manuscript. As depicted in Figure R6a, with the increase in PVCA content, the peak intensity at 18.2° gradually diminishes, while the characteristic peak at approximately 21.1° exhibits enhanced intensity, indicating an increase in crystallinity. The crystallinity of this polymer increases with the increase of PVCA content, indicating that the intermolecular interaction between PVCA and PVFH increases⁴⁻⁶.

As for how to prepare XRD samples, in order to ensure that the polymer dry film with different components has the same thickness, we completely dissolved PVFH and PVCA of different proportions in DMSO to form a mixed solution with the same mass fraction, injected the above solution with the same mass into the same type of PTFE mold, and dried it under vacuum at 80°C for 48h. The fully dried polymer films were cut into 16mm diameter discs, and the different polymer films with the same thickness were placed on the same sample table for testing under the same ambient temperature and humidity.

Figure R6. (a) PVFH-PVCA without liquid electrolyte XRD of different PVCA contents. (b) XRD spectra of PVCA, PVFH, PVFH-PVCA without LE and PVFH-PVCA, respectively.

Changes made in the revised manuscript:

We have revised **Supplementary Fig. 5a** and updated the corresponding discussion in the first paragraph of the **Structure and morphology of PVFH-PVCA blends** section, as follows: “For mixtures of PVCA and PVFH, with the increase in PVCA content, the peak intensity at 18.2° gradually diminishes, while the characteristic peak at approximately 21.1° exhibits enhanced intensity, This suggests that with the increasing PVCA content, the intermolecular interactions between PVCA and PVFH strengthen, leading to a modification in the polymer's crystallinity. When the PVCA content reaches 15%, the characteristic peak at 18.2° vanishes, and the ~21.1° characteristic peak not only shifts to ~21.8° but also exhibits an increased intensity. This indicates that the entanglement interactions between the two components have augmented the polymer's crystallinity²⁹⁻³¹ (Supplementary Fig. S5a).”

Supplementary Fig. 5 (a) PVFH-PVCA without liquid electrolyte XRD of different PVCA contents. (b) XRD spectra of PVCA, PVFH, PVFH-PVCA without LE and PVFH-PVCA, respectively.

Comment 4. The modulus data from tensile and AFM are off by nearly three orders of magnitudes.

Response: Thanks for reviewer's question. The substantial disparity in modulus data between tensile testing and atomic force microscopy testing may be attributed to the following factors:

1. **Scale Effect:** Tensile testing is typically employed for materials at the macroscopic scale, whereas atomic force microscopy testing operates at the microscopic or even nanoscale. At the microscopic level, the atomic and molecular structure of materials begins to play a significant role, potentially resulting in different mechanical properties compared to the macroscopic scale. Atomic force microscopy measurements are often performed closer to the material's surface, while tensile testing encompasses the entire bulk material. This difference in scale could lead to significant disparities in modulus data.
2. **Surface Effect:** Atomic force microscopy testing measures interaction forces at the material's surface, whereas tensile testing quantifies the relationship between force and displacement throughout the entire sample. Material surfaces typically exhibit different structures and properties, which may render them more brittle or harder, impacting modulus values. Consequently, atomic force microscopy testing may yield relatively higher modulus values.
3. **Experimental Conditions:** Different testing methods are conducted under varying experimental conditions, including factors such as the strain rate, ambient temperature, and humidity. These conditions can have a significant impact on the mechanical properties of the materials. The tensile testing was performed under normal room conditions at 30°C, which resulted in modulus values lower than those obtained under ambient temperature and dry conditions as measured by AFM.

In conclusion, the significant disparities arise from the fact that these two testing methods pertain to different scales and surface effect. Similar experimental results have been reported in previous literature as well. For instance, Kang et al⁷. reported an organic-inorganic composite electrolyte with an average Young's modulus of 1.22 GPa and a fracture toughness of 2.12 MPa. On the other hand, Xu et al⁸. reported a Young's modulus as high as 12.6 GPa for AO-PIM-1-Li membranes, while the modulus obtained from tensile tests for AO-PIM-1-Li was 50.0 MPa.

Therefore, it should be pointed out that when measuring modulus, the corresponding experimental methods and conditions must be specified.

Comment 5. Figure 3a shows that the oxidation potential of PVFH-PVCA is also lower than 5 V, as there is a small step before 5V.

Response: Thanks for reviewer's careful check. Indeed, there is a minor oxidation peak at 5.03V in the initial manuscript. According to some previous works^{9,10}, when the specific current in the electrochemical stability window measurement by LSV method increases exponentially, the electrochemical stability window was determined using tangent intersection method, resulting in an electrochemical stability window of 5.3V. Owing to the slow oxidation process of electrolytes, the impact of changes in scanning rate is significant. The presence of multiple variables in LSV measurement poses a challenge in comparing and accurately evaluating the stability of different electrolyte systems. Therefore, LSV testing methods with appropriate scanning rates are only

suitable for providing qualitative comparisons between different electrolytes. In the revised manuscript, all the scanning speed is set at 1mv/s. and this minor oxidation peak did not appear again during repeated measurements. Electrochemical floating experiments is more promising and accurate in evaluating electrochemical stability windows. Owing to the long-term constant voltage condition ensuring sufficient reaction time, the measured electrochemical stability window is not affected by the scanning rate. In present work, before evaluating the battery performance, electrochemical floating experiments were conducted on the cells with PVFH-PVCA to accurately determine their actual operational electrochemical windows (Supplementary Fig. S27). For the PVFH-PVCA, the measured leakage current was less than 20 μA up to a voltage of 4.9 V.

Changes made in the revised manuscript:

After three repeated measurements, Fig.3a was changes in the revised manuscript.

Fig. 3 | Properties of the PVFH-PVCA electrolyte. a, Electrochemical stability window of PVFH-PVCA, CLE, and PVLE evaluated by linear sweep voltammetry with a scan rate of 1 mV/s

Comment 6. Figure 5a, are the samples mislabeled? The figure shows CLE has the longest cycling life.

Response: Thanks for reviewer' careful check. We have revised the Figure.5a in the revised manuscript.

Changes made in the revised manuscript:

Fig.5a was revised as follows:

Fig. 5 | Characterization of Li || Li symmetric cells. a, Galvanostatic voltage profiles of Li||Li symmetric cells with different electrolytes at 8.0 mA cm⁻² and 4.0 mAh cm⁻².

References:

- 1 Avella, A., Mincheva, R., Raquez, J.-M. & Lo Re, G. Substantial Effect of Water on Radical Melt Crosslinking and Rheological Properties of Poly(ϵ -Caprolactone). *Polymers* **13**, 491 (2021).
- 2 Pei, F. *et al.* Titanium–oxo cluster reinforced gel polymer electrolyte enabling lithium–sulfur batteries with high gravimetric energy densities. *Energy & Environmental Science* **14**, 975–985, doi:10.1039/D0EE03005H (2021).
- 3 Shim, J. *et al.* 2D boron nitride nanoflakes as a multifunctional additive in gel polymer electrolytes for safe, long cycle life and high rate lithium metal batteries. *Energy & Environmental Science* **10**, 1911–1916, doi:10.1039/C7EE01095H (2017).
- 4 Zou, L. & Zhang, W. Molecular Dynamics Simulations of the Effects of Entanglement on Polymer Crystal Nucleation. *Macromolecules* **55**, 4899–4906, doi:10.1021/acs.macromol.2c00817 (2022).
- 5 Zhu, C. & Zhao, J. Nucleation and Crystallization of Polymer Melts under Cyclic Stretching: Entanglement Effect. *Macromolecules* **56**, 5490–5501, doi:10.1021/acs.macromol.3c00461 (2023).
- 6 Ni, L. *et al.* Retarded Crystallization and Promoted Phase Transition of Freeze-Dried Polybutene-1: Direct Evidence for the Critical Role of Chain Entanglement. *ACS Macro Letters* **11**, 257–263, doi:10.1021/acsmacrolett.1c00794 (2022).
- 7 Kang, J. *et al.* Heterojunction-Accelerating Lithium Salt Dissociation in Polymer Solid Electrolytes. *Advanced Functional Materials* **n/a**, 2307263, doi:<https://doi.org/10.1002/adfm.202307263>.
- 8 Wang, X.-X. *et al.* Polymers with Intrinsic Microporosity as Solid Ion Conductors for Solid-State Lithium Batteries. *Angewandte Chemie International Edition* **62**, e202308837, doi:<https://doi.org/10.1002/anie.202308837> (2023).
- 9 Kasnatscheew, J. *et al.* Determining oxidative stability of battery electrolytes: validity of common electrochemical stability window (ESW) data and alternative strategies. *Physical Chemistry Chemical Physics* **19**, 16078–16086, doi:10.1039/C7CP03072J (2017).
- 10 Méry, A., Rousselot, S., Lepage, D. & Dollé, M. A Critical Review for an Accurate Electrochemical Stability Window Measurement of Solid Polymer and Composite Electrolytes. *Materials* **14**, 3840 (2021).

REVIEWER COMMENTS

Reviewer #2 (Remarks to the Author):

The revised manuscript addressed some of the original concerns, while some questions remain unclear.

1) In response to the original comment 3 on WAXD, the author mentioned, "Thanks for the reviewer's question. Figure R6a was normalized in the revised manuscript. As depicted in Figure R6a, with the increase in PVCA content, the peak intensity at 18.2° gradually diminishes, while the characteristic peak at approximately 21.1° exhibits enhanced intensity, indicating an increase in crystallinity. The crystallinity of this polymer increases with the increase of PVCA content, indicating that the intermolecular interaction between PVCA and PVFH increases". What are the 18.2 and 21.1 peaks corresponding to? How do you tell the crystallinity increase? Please estimate the crystallinity. Why would the increase in crystallinity indicate enhanced intermolecular interaction? One would argue that stronger intermolecular interaction between dissimilar polymers could interfere with chain packing, leading to lower crystallinity.

2) In response to the original comment 3 on mechanical measurement. This review cannot agree that different measurement methods of the moduli could lead to 3 orders of magnitude difference. The three reasons that the authors suggested are qualitative at best and does not explain 3 orders of magnitude difference. How was the AFM calibrated? For a typical polymer film, the moduli measured by tensile and AFM are quite close.

Reviewer #3 (Remarks to the Author):

As I am an adjudicating reviewer after an initial round of reviews, my comments will be brief. It appears to me that the authors have been very responsive to the prior reviewers, and if the reviewers are happy with the revisions (presumably they will be asked to review the revised paper), then that's ok with me too. However, I do need to point out an obvious error in the first sentence of the Material Characterization section: ^{13}C NMR spectroscopy – not ^{12}C ! Perhaps less obvious to the general reader but very obvious to NMR people, I do not believe the T2 results in Fig. 2a – you will not detect a spin echo in a solid or quasi-solid polymer with a 60 sec pulse separation. The time unit is clearly mislabeled (could be μs or ms, at most).

Response to reviewers

We gratefully thank all the reviewers for their time spent making their constructive remarks and useful suggestions, which can enable us to make improvements and significantly raise the quality of the manuscript. Each suggested revision and comment, brought forward by the reviewers was accurately incorporated and considered. Below the comments of the reviewers are addressed point-by-point and the revisions are indicated.

Response to reviewer's comments

Reviewer #2 (Remarks to the Author):

The revised manuscript addressed some of the original concerns, while some questions remain unclear.

Comment 1: In response to the original comment 3 on WAXD, the author mentioned, "Thanks for the reviewer's question. Figure R6a was normalized in the revised manuscript. As depicted in Figure R6a, with the increase in PVCA content, the peak intensity at 18.2° gradually diminishes, while the characteristic peak at approximately 21.1° exhibits enhanced intensity, indicating an increase in crystallinity. The crystallinity of this polymer increases with the increase of PVCA content, indicating that the intermolecular interaction between PVCA and PVFH increases". What are the 18.2 and 21.1 peaks corresponding to? How do you tell the crystallinity increase? Please estimate the crystallinity. Why would the increase in crystallinity indicate enhanced intermolecular interaction? One would argue that stronger intermolecular interaction between dissimilar polymers could interfere with chain packing, leading to lower crystallinity.

Response: Thank you for reviewer's professional questions.

i) The peak at 17.4° can be assigned to the contribution of the pure PVCA (**Figure R1a**). Two peaks at 18.2° and 21.1° can be assigned to the contribution of the pure PVFH (**Figure R1a**).

ii) How do you tell the crystallinity increase? In XRD analysis, crystallinity can be determined by calculating the intensity and shape of the diffraction peaks. The greater the number of crystals in the sample, the higher the intensity of the diffraction peaks. Additionally, when the crystals are well-arranged, the diffraction peaks appear sharper and narrower, indicating higher crystallinity. Crystallinity can be quantified by the shape of the diffraction peaks. The most commonly used method is to represent crystallinity by half the full width at half peak (FWHM); the smaller the FWHM, the higher the crystallinity. Another method is using the ratio of the maximum peak intensity to the minimum peak intensity; the higher the ratio, the higher the crystallinity.

In present work, from **Figure R.1a**, when PVCA and PVFH begin to mix, due to the proximity of PVCA's characteristic peak (17.4°) and one of PVFH's characteristic peaks (18.2°), a broad peak appears around $\sim 18^\circ$. With an increase in PVCA content (PVFH content decreases), the intensity of the characteristic peak ($\sim 18^\circ$) gradually decreases, while the PVFH's characteristic peak at 21.1° maintains a sharp shape and sufficiently high intensity, indicating an overall improvement in the electrolyte's crystallinity, primarily contributed by PVFH.

iii) In present work, why would the increase in crystallinity indicate enhanced intermolecular interaction? Textbooks in polymer physics [1] suggested that the crystallinity of polymers represents the segmental order of chains. Factors influencing the segmental order mainly include segment flexibility, length, and inter-segmental forces. The stronger the inter-segmental forces, the greater the likelihood of ordered segment arrangement and higher crystallinity. In the PVFH-PVCA system, the addition of PVCA leads to strong inter-chain interactions, which favor the crystallization of the polymer component PVFH^{1,2}. This occurs due to the extended polymer chain conformation, resulting in dense chain stacking and increased PVFH crystallization, as similarly reported in blend systems of these two polymers³. For the shorter-chain PVCA with smaller molecular weight,

stronger intermolecular interactions interfere with the linear arrangement of PVCA. PVCA molecules are intertwined by the larger PVFH polymer chains, affecting the crystallinity of PVCA, leading to a decrease in crystallinity, manifested by the disappearance of XRD characteristic peaks, a common occurrence when introducing small-molecule polymers into polymer networks. Polymer networks tend to inhibit the aggregation of small molecules. Due to the differences in relative molecular weights, the receptor's crystallization is usually asymmetric, with weak crystallization of small-molecule receptors, tending toward amorphous states or forming small-sized crystals in mixed systems^{4,5}.

Figure R1. (a) PVFH-PVCA without liquid electrolyte XRD of different PVCA contents. (b) XRD spectra of PVCA, PVFH, PVFH-PVCA without LE and PVFH-PVCA, respectively.

Changes made in the revised manuscript:

We have revised **Supplementary Fig. 5a** and updated the corresponding discussion in the first paragraph of the **Structure and morphology of PVFH-PVCA blends** section, as follows: "For mixtures of PVCA and PVFH, the addition of PVCA leads to strong inter-chain interaction, which favor the crystallization of the polymer PVFH with larger molecular weight. This occurs due to the extended polymer chain conformation, resulting in dense chain stacking and increased PVFH crystallization, as similarly reported in blend systems of these two polymers²⁹. However, the stronger intermolecular interaction interferes with the linear arrangement of the shorter-chain PVCA with smaller molecular weight. And PVCA molecules are intertwined by the larger PVFH polymer chains, affecting the crystallinity of PVCA, leading to a decrease in crystallinity^{30,31}, manifested by the gradually disappearance of the characteristic peak at $\sim 18^\circ$ with the increase in PVCA content (Supplementary Fig. S5a)."

Supplementary Fig. 5 | (a) PVFH-PVCA without liquid electrolyte XRD of different PVCA contents. (b) XRD spectra of PVCA, PVFH, PVFH-PVCA without LE and PVFH-PVCA, respectively.

Comment 2: In response to the original comment 3 on mechanical measurement. This review cannot agree that different measurement methods of the moduli could lead to 3 orders of magnitude difference. The three reasons that the authors suggested are qualitative at best and does not explain 3 orders of magnitude difference. How was the AFM calibrated? For a typical polymer film, the moduli measured by tensile and AFM are quite close.

Response: Thanks for reviewer's careful check. We meticulously examined the previous testing process and identified the source of the error. In the previous tests, we used the DNISP-HS probe with a range of $10 \text{ GPa} < E < 100 \text{ GPa}$. This range was excessive, and during the testing process, the probe came into contact with the substrate, leading to inflated test results. In the revised manuscript, we conducted AFM modulus testing again, and based on the modulus of the tensile machine, we selected the RTESPA-150 probe with a range of $5 \text{ MPa} < E < 500 \text{ MPa}$. The calibration method employed was a relative method. The specific steps involved first measuring the mechanical curve of a diamond sample to establish the relationship between the probe's indentation force and its deflection. Next, we calibrated the modulus of a standard sample, PDMS-SOFT-2-12M. Finally, the PVFH-PVCA and PVLE samples are measured. For the PVFH-PVCA sample, we tested four locations to ensure data accuracy. We employed the same method to re-measure and collect data for the control sample PVLE. The experimental results for PVFH-PVCA are as follows:

Figure R2. Shear modulus of PVFH-PVCA.

Changes made in the revised manuscript:

We have revised **Fig. 2d** and **Supplementary Fig. 15c** and updated the corresponding discussion in the second paragraph of the **Mechanical and electrochemical properties of PVFH-PVCA** section, as follows: “Atomic force microscopy (AFM) investigations revealed that the PVFH-PVCA membrane has a significantly lower surface roughness and a much larger Young's modulus(188.5MPa) than PVLE (26.5MPa, Fig. 2d and Supplementary Fig. S15).”

Reviewer #3 (Remarks to the Author):

As I am an adjudicating reviewer after an initial round of reviews, my comments will be brief. It appears to me that the authors have been very responsive to the prior reviewers, and if the reviewers are happy with the revisions (presumably they will be asked to review the revised paper), then that's ok with me too. However, I do need to point out an obvious error in the first sentence of the Material Characterization section: ^{13}C NMR spectroscopy – not ^{12}C ! Perhaps less obvious to the general reader but very obvious to NMR people, I do not believe the T2 results in Fig. 2a – you will not detect a spin echo in a solid or quasi-solid polymer with a 60 sec pulse separation. The time unit is clearly mislabeled (could be μs or ms, at most).

Response: Thanks for reviewer's careful check. In the revised manuscript, we changed ^{12}C to ^{13}C in "**Material characterisation**". By checking the original data, we discovered that the time unit in Fig 2a is "ms" rather than "s". This mistake was due to our oversight, and as a result, we have corrected the time unit. Once again, we thank the reviewer for helping us improve the quality of the paper.

Changes made in the revised manuscript:

We have corrected ^{12}C to ^{13}C in the "**Material characterisation**". We have made amendments to **Fig. 2a** and revised the discussion concerning **Fig. 2a**, as shown below: "Similarly, the relaxation curve of PVFH-PVCA indicated rapid decay times (less than ~20 ms), corresponding to the motion of entangled chains within the system, and slow decay times (greater than ~30 ms), representing the motion of chain ends and short chains within the system."

Fig.2a was revised as follows:

- 1 Klonos, P. A. *et al.* Molecular mobility, crystallization and melt-memory investigation of molar mass effects on linear and hydroxyl-terminated Poly(ϵ -caprolactone). *Polymer* **242**, 124603, doi:<https://doi.org/10.1016/j.polymer.2022.124603> (2022).
 - 2 Yuan, D. *et al.* Synergy between Photoluminescence and Charge Transport Achieved by Finely Tuning Polymeric Backbones for Efficient Light-Emitting Transistor. *Journal of the American Chemical Society* **143**, 5239-5246, doi:10.1021/jacs.1c01659 (2021).
 - 3 Zhang, Q. *et al.* High-temperature polymers with record-high breakdown strength enabled by rationally designed chain-packing behavior in blends. *Matter* **4**, 2448-2459, doi:<https://doi.org/10.1016/j.matt.2021.04.026> (2021).
 - 4 Howard, I. A., Mauer, R., Meister, M. & Laquai, F. Effect of Morphology on Ultrafast Free Carrier Generation in Polythiophene:Fullerene Organic Solar Cells. *Journal of the American Chemical Society* **132**, 14866-14876, doi:10.1021/ja105260d (2010).
 - 5 Ye, L. *et al.* High-Efficiency Nonfullerene Organic Solar Cells: Critical Factors that Affect Complex Multi-Length Scale Morphology and Device Performance. *Advanced Energy Materials* **7**, 1602000, doi:<https://doi.org/10.1002/aenm.201602000> (2017).
- [1] Rubinsten M. Polymer physics[M]. United States of America, 2003.

Reviewers' comments:

Reviewer #2 (Remarks to the Author):

The AFM results were improved. However, this reviewer cannot agree on the explanations regarding the XRD results.

1. The authors stated that "How do you tell the crystallinity increase? In XRD analysis, ... Crystallinity can be quantified by the shape of the diffraction peaks. The most commonly used method is to represent crystallinity by half the full width at half peak (FWHM); the smaller the FWHM, the higher the crystallinity." This is wrong. FWHM is related to the correlation length, which could be completely unrelated to crystallinity.
2. "...Another method is using the ratio of the maximum peak intensity to the minimum peak intensity; the higher the ratio, the higher the crystallinity." What does this mean? Can the authors, based on this method, calculate the crystallinity of the samples they show?
3. "...In present work, from Figure R.1a, when PVCA and PVFH begin to mix, due to the proximity of PVCA's characteristic peak (17.4°) and one of PVFH's characteristic peaks (18.2°), a broad peak appears around $\sim 18^\circ$. With an increase in PVCA content (PVFH content decreases), the intensity of the characteristic peak ($\sim 18^\circ$) gradually decreases, while the PVFH's characteristic peak at 21.1° maintains a sharp shape and sufficiently high intensity, indicating an overall improvement in the electrolyte's crystallinity, primarily contributed by PVFH." This is also unclear. What are the numeric values of the crystallinity? When discussion crystallinity change, numeric values are needed. Again, please note that a sharp peak does not mean high crystallinity.
4. "In present work, why would the increase in crystallinity indicate enhanced intermolecular interaction? Textbooks in polymer physics [1] suggested that the crystallinity of polymers represents the segmental order of chains. ... The stronger the inter-segmental forces, the greater the likelihood of ordered segment arrangement and higher crystallinity. ..." This is also wrong. For example, polyethylene is nonpolar and has weak intersegmental forces, but linear polyethylene has high crystallinity and often exceeds 90%.

Response to reviewer's comments

Comment 1. The authors stated that “How do you tell the crystallinity increase? In XRD analysis, ... Crystallinity can be quantified by the shape of the diffraction peaks. The most commonly used method is to represent crystallinity by half the full width at half peak (FWHM); the smaller the FWHM, the higher the crystallinity.” This is wrong. FWHM is related to the correlation length, which could be completely unrelated to crystallinity.

Response: Thank you for your professional questions. When the particles are in an amorphous state, the XRD peak widens into a diffuse peak, and FWHM is significantly larger than that of well crystallized samples. Therefore, it can be qualitatively assumed that FWHM is related to crystallinity¹. This rule is common in material crystallography and cannot be considered incorrect. For example, in the study of polyethylene and polypropylene blends², the author proposed that "The HDPE/PP and LLDPE/PP blends show sharper X-ray diffraction patterns than the MEPE/PP blends. The FWHM of HDPE/PP and LLDPE/PP blends changes only slightly with a variation in the blend ratio but that of MEPE/PP changes significantly. This suggests that MEPE affects the crystallization of PP in the MEPE/PP blends". In the study of π -Conjugated organic molecules³, the authors proposed that “All the polymers show typical face-on dominated molecular packing in solid state. p(TDPP-BT) shows the narrower full width at half maximum (FWHM) with a lamellar distance of 21.67 Å and a π - π stacking distance of 3.57 Å, suggesting its higher ordered crystallite domain”. In the study of thin film materials⁴, the authors proposed “In an X-ray pattern, the broadening of a peak can be due to different factors. The peak broadening can be due to the broader distribution of tiny or bigger-sized particles. It can be due to the porosity factor. The broadening of a peak can be due to a non-flat surface at the atomic level. Under such factors, it keeps low intensity. It can also show bars and turns. The FWHM of the peak can be more”. These works of polymer and thin film materials indicate the relationship between FWHM of XRD peaks and crystallinity. In fact, it is easy to find more support work in other materials fields⁵⁻⁹. It should also be pointed out that the Scherrer equation^[1] has been used to calculate the coherence length or grain size, and the results showed that the coherence length is proportional to $1/\text{FWHM}$ ¹⁰.

Inspired by reviewer's questions, we have checked the standard definition of crystallinity and realized the misunderstanding of crystallinity and the degree of long-range order. To avoid misunderstandings and improve the readability of the manuscript, we have decided to completely remove XRD analysis and related discussions from the article. Removing the content of XRD characterization will not have any impact on the scientific and readability of the article due to the fact that XRD characterization is not the focus of our work and is not related to other studies.

Changes made in the revised manuscript:

We have completely removed the XRD analysis and related discussion in the revised manuscript. The paragraph: “For mixtures of PVCA and PVFH, the addition of PVCA leads to strong inter-chain interaction, which favors the ordered arrangement of the polymer PVFH with larger molecular weight. This occurs due to the extended polymer chain conformation, resulting in dense chain stacking and increased PVFH crystallization, as similarly reported in blend systems of these two

polymers²⁹. Meanwhile, the stronger intermolecular interaction interferes with the linear arrangement of the shorter-chain PVCA^{30,31}, manifested by the gradually disappearance of the characteristic peak at ~18° with the increase in PVCA content (Supplementary Fig. S5a). This result indicates a strong interaction between PVCA and the PVFH matrix. In contrast, only a broad peak appeared at ~21° for the PVFH-PVCA membrane, indicating its highly amorphous state (Supplementary Fig. S5b).” was removed in the **Structure and morphology of PVFH-PVCA blends** section.

Comment 2. “...Another method is using the ratio of the maximum peak intensity to the minimum peak intensity; the higher the ratio, the higher the crystallinity.” What does this mean? Can the authors, based on this method, calculate the crystallinity of the samples they show?

Response: In the previous response, in order to answer the reviewer's question about the determination method of sample crystallinity in previous literatures and to make the response more readable, we introduced the method of using the ratio of maximum peak intensity to minimum peak intensity to explain crystallinity. In fact, we did not use this method in our work.

It should be noted that using the ratio of maximum peak intensity to minimum peak intensity to preliminarily determine crystallinity is a qualitative and convenient method, especially suitable for studying a series of polymers with the same composition but different contents¹¹. In the study of artificial silk yarn and cellulose raw materials, Conrad et al¹². proposed this method and calculated the crystallinity using this equation:

$$\text{Crystallinity} = \frac{I_{(101)} - I_{\text{Min}}}{I_{(101)}} \times 100\%$$

$I_{(101)}$ reports the intensity of (101) peak. Similar methods have also been proposed by works from Matthews et al¹³, and Clark et al¹⁴. Wakelin et al.^{15,16} suggested that the degree of crystallinity can be qualitatively judged by Correlation Crystallinity Index. A simple way to use this index is to calculate the absolute difference between the intensity of crystalline peak (I_c) and amorphous peak (I_a). In summary, the difference or ratio between I_{max} and I_{min} represents the difference in crystallinity between crystalline and amorphous materials. The larger the difference, the higher the crystallinity. It should be emphasized again that we did not use the method in our work. To avoid the misunderstanding, we have completely removed the XRD analysis and associated discussion from the manuscript.

Comment 3. ”In present work, from Figure R.1a, when PVCA and PVFH begin to mix, due to the proximity of PVCA's characteristic peak (17.4°) and one of PVFH's characteristic peaks (18.2°), a broad peak appears around ~18°. With an increase in PVCA content (PVFH content decreases), the intensity of the characteristic peak (~18°) gradually decreases, while the PVFH's characteristic peak at 21.1° maintains a sharp shape and sufficiently high intensity, indicating an overall improvement in the electrolyte's crystallinity, primarily contributed by PVFH.” This is also unclear. What are the numeric values of the crystallinity? When discussion crystallinity change, numeric values are needed. Again, please note that a sharp peak does not mean high crystallinity.

Response: In the XRD measurement of inorganic nanoparticles or organic polymers, if the peak is strong and sharp, it means that the inorganic nanoparticles are larger^{17,18,[1]}, while in polymers, the strength of crystallinity can also be qualitatively determined by the sharpness of the peak shape¹⁹. The sharpness of characteristic peaks in XRD is related to the size of nanoparticles, and the larger the particle, the sharper the characteristic peak (note that nanoparticles should be less than 100nm^{20,[2]}); On the other hand, it is also related to crystallinity. In high polymers, such as polyethylene, high-density polyethylene has higher crystallinity than low-density polyethylene, with stronger and sharper crystalline diffraction peaks^{21,22}. It is worth noting that the PVFH-PVCA we measured is a polymer film, without the presence of powder nanoparticles. Under the same measurement conditions and polymer film thickness, the crystallinity can be qualitatively determined by the sharpness and intensity of characteristic peaks after normalization. The characteristic peak of PVFH at ~21.1 ° maintains a sharp shape and sufficiently high intensity, indicating an overall improvement in electrolyte crystallinity, mainly due to the contribution of PVFH. This can be qualitatively determined by the changes in the characteristic peak of XRD. To avoid the misunderstanding, we have completely removed the XRD analysis and associated discussion from the manuscript.

Comment 4. “In present work, why would the increase in crystallinity indicate enhanced intermolecular interaction? Textbooks in polymer physics [1] suggested that the crystallinity of polymers represents the segmental order of chains. ... The stronger the inter-segmental forces, the greater the likelihood of ordered segment arrangement and higher crystallinity. ...” This is also wrong. For example, polyethylene is nonpolar and has weak intersegmental forces, but linear polyethylene has high crystallinity and often exceeds 90%.

Response: In polymer crystals, the c-direction of the crystal cell is the central axis direction of the polymer main chain, and a long polymer chain needs to pass through multiple crystal cells. Therefore, only when the polymer chain itself has a high degree of spatial structural regularity can it achieve remote ordered arrangement in the c-direction and have crystallization ability. However, polymers with irregular chain structures lack crystallization ability. The relationship between the regularity of polymer chain structure and crystallization ability is as follows:²³

(1) Symmetry of chains

The higher the symmetry of the polymer chain structure, the stronger the crystallization ability. For example, polyethylene with simple and highly symmetrical structural units has extremely strong crystallization ability.

(2) The spatial regularity of chains

The spatial conformation/conformational regularity of chains affects the crystallization ability of polymers. Organized polymers obtained through directed polymerization may crystallize.

(3) Polymers have intermolecular forces, such as hydrogen bonds, which are beneficial for improving crystallization ability.

As the reviewer pointed out, the intermolecular forces between linear polyethylene segments are weak, but the crystallinity is high. This is because linear polyethylene has good structural unit symmetry, no branching chains, and high molecular chain order, so the crystallinity of polyethylene

is strong ^[3]. The example provided by the reviewer is consistent with the relationship in (1). And the orderliness of segments is related not only to the symmetry of the chain, but also to other factors, as stated in the book, "Essential conditions for crystallization are the symmetry of the structural units and the strength of the interactive forces between different molecules."²³ The strong intermolecular forces between polymers are beneficial for improving segment order and thus enhancing crystallization ability, which is consistent with the relationship in (3). In our work, there is a strong interaction force between the PVFH-PVCA molecular chains, which promotes the orderliness of the chain structure. So it is not incorrect to say that the stronger the force between segments, the greater the possibility of orderly arrangement of segments, and the higher the crystallinity. However, the applicable conditions are different.

To avoid the misunderstanding, we have completely removed the XRD analysis and associated discussion from the manuscript.

References

- 1 Murthy, N. S. & Minor, H. General procedure for evaluating amorphous scattering and crystallinity from X-ray diffraction scans of semicrystalline polymers. *Polymer* **31**, 996-1002, doi:[https://doi.org/10.1016/0032-3861\(90\)90243-R](https://doi.org/10.1016/0032-3861(90)90243-R) (1990).
- 2 Furukawa, T. *et al.* Molecular Structure, Crystallinity and Morphology of Polyethylene/Polypropylene Blends Studied by Raman Mapping, Scanning Electron Microscopy, Wide Angle X-Ray Diffraction, and Differential Scanning Calorimetry. *Polymer Journal* **38**, 1127-1136, doi:10.1295/polymj.PJ2006056 (2006).
- 3 Chen, X.-X. *et al.* High-mobility semiconducting polymers with different spin ground states. *Nature Communications* **13**, 2258, doi:10.1038/s41467-022-29918-w (2022).
- 4 Ali, M. Qualitative analyses of thin film-based materials validating new structures of atoms. *Materials Today Communications* **36**, 106552, doi:<https://doi.org/10.1016/j.mtcomm.2023.106552> (2023).
- 5 Pergolesi, D. *et al.* High proton conduction in grain-boundary-free yttrium-doped barium zirconate films grown by pulsed laser deposition. *Nature Materials* **9**, 846-852, doi:10.1038/nmat2837 (2010).
- 6 Poccia, N. *et al.* Evolution and control of oxygen order in a cuprate superconductor. *Nature Materials* **10**, 733-736, doi:10.1038/nmat3088 (2011).
- 7 Hayashi, A., Noi, K., Tanibata, N., Nagao, M. & Tatsumisago, M. High sodium ion conductivity of glass-ceramic electrolytes with cubic Na₃PS₄. *Journal of Power Sources* **258**, 420-423, doi:<https://doi.org/10.1016/j.jpowsour.2014.02.054> (2014).
- 8 Makiura, R. *et al.* Surface nano-architecture of a metal-organic framework. *Nature Materials* **9**, 565-571, doi:10.1038/nmat2769 (2010).
- 9 Li, X. *et al.* Facile transformation of imine covalent organic frameworks into ultrastable crystalline porous aromatic frameworks. *Nature Communications* **9**, 2998, doi:10.1038/s41467-018-05462-4 (2018).
- 10 Patterson, A. L. The Scherrer Formula for X-Ray Particle Size Determination. *Physical Review* **56**, 978-982, doi:10.1103/PhysRev.56.978 (1939).
- 11 Uzun, İ. Methods of determining the degree of crystallinity of polymers with X-ray diffraction: a review. *Journal of Polymer Research* **30**, 394, doi:10.1007/s10965-023-03744-0 (2023).
- 12 Conrad, C. C. & Scroggie, A. G. Chemical Characterization of Rayon Yarns and Cellulosic Raw Materials. *Industrial & Engineering Chemistry* **37**, 592-598, doi:10.1021/ie50426a025 (1945).
- 13 Matthews, J. L., Peiser, H. S. & Richards, R. B. THE X-RAY MEASUREMENT OF THE AMORPHOUS CONTENT OF POLYTHENE SAMPLES. *Acta Crystallographica* **2**, 85-&, doi:10.1107/s0365110x49000199 (1949).
- 14 Clark, G. L. & Terford, H. C. Quantitative X-Ray Determination of Amorphous Phase in Wood Pulps as Related to Physical and Chemical Properties. *Analytical Chemistry* **27**, 888-895, doi:10.1021/ac60102a006 (1955).
- 15 Wakelin, J. H., Virgin, H. S. & Crystal, E. Development and Comparison of Two X-Ray Methods for Determining the Crystallinity of Cotton Cellulose. *Journal of Applied Physics* **30**, 1654-1662, doi:10.1063/1.1735031 (2004).
- 16 Statton, W. O. An X-ray crystallinity index method with application to poly(ethylene terephthalate). *Journal of Applied Polymer Science* **7**, 803-815, doi:<https://doi.org/10.1002/app.1963.070070302> (1963).

- 17 Feng, G. *et al.* Sub-2 nm Ultrasmall High-Entropy Alloy Nanoparticles for Extremely Superior Electrocatalytic Hydrogen Evolution. *Journal of the American Chemical Society* **143**, 17117-17127, doi:10.1021/jacs.1c07643 (2021).
- 18 Uvarov, V. & Popov, I. Metrological characterization of X-ray diffraction methods for determination of crystallite size in nano-scale materials. *Materials Characterization* **58**, 883-891, doi:<https://doi.org/10.1016/j.matchar.2006.09.002> (2007).
- 19 Li, D., Zhou, L., Wang, X., He, L. & Yang, X. Effect of Crystallinity of Polyethylene with Different Densities on Breakdown Strength and Conductance Property. *Materials* **12**, 1746 (2019).
- 20 Hashemi-Sadraei, L., Mousavi, S. E., Lavernia, E. J. & Schoenung, J. M. The Influence of Grain Size Determination Method on Grain Growth Kinetics Analysis. *Advanced Engineering Materials* **17**, 1598-1607, doi:<https://doi.org/10.1002/adem.201500057> (2015).
- 21 Zhou, L., Wang, X., Zhang, Y., Zhang, P. & Li, Z. An Experimental Study of the Crystallinity of Different Density Polyethylenes on the Breakdown Characteristics and the Conductance Mechanism Transformation under High Electric Field. *Materials* **12**, 2657 (2019).
- 22 Oliveira, A. D. B. *et al.* HDPE/LLDPE blends: rheological, thermal, and mechanical properties. *Materials Research Innovations* **24**, 289-294, doi:10.1080/14328917.2019.1655623 (2020).
- 23 Münstedt, H. & Schwarzl, F. R. in *Deformation and Flow of Polymeric Materials* (eds Helmut Münstedt & Friedrich Rudolf Schwarzl) 77-120 (Springer Berlin Heidelberg, 2014).

[1] Scherrer, P. (1918) Estimation of the Size and Internal Structure of Colloidal Particles by Means of Röntgen. *Nachrichten von der Gesellschaft der Wissenschaften zu Göttingen*, 2, 96-100.

[2] H. P. Klug, L. E. Alexander, in *X-ray Diffraction Procedures for Polycrystalline and Amorphous Materials*, John Wiley & Sons, New York 1974, p. 618.

[3] Peacock, A. (2000). *Handbook of Polyethylene: Structures: Properties, and Applications* (1st ed.). CRC Press. <https://doi.org/10.1201/9781482295467>